# PDIA1/P4HB is required for efficient proinsulin maturation and ß cell health in response to diet induced obesity

Insook Jang[1], Anita Pottekat[1], Juthakorn Poothong[1], Jing Yong[1], Jacqueline Lagunas-Acosta[1], Adriana Charbono[1], Zhouji Chen[1], Donalyn L Scheuner[2], Ming Liu[3], Pamela Itkin-Ansari[4], Peter Arvan[3]*, Randal J Kaufman[1]*

[1]Degenerative Diseases Program, SBP Medical Discovery Institute, La Jolla, United States; [2]DLS Consulting, Greenfield, United States; [3]Division of Metabolism Endocrinology and Diabetes, University of Michigan Medical School, Ann Arbor, United States; [4]Department of Pediatrics, University of California, San Diego, San Diego, United States

**Abstract** Regulated proinsulin biosynthesis, disulfide bond formation and ER redox homeostasis are essential to prevent Type two diabetes. In ß cells, protein disulfide isomerase A1 (PDIA1/ *P4HB*), the most abundant ER oxidoreductase of over 17 members, can interact with proinsulin to influence disulfide maturation. Here we find *Pdia1* is required for optimal insulin production under metabolic stress in vivo. ß cell-specific *Pdia1* deletion in young high-fat diet fed mice or aged mice exacerbated glucose intolerance with inadequate insulinemia and increased the proinsulin/insulin ratio in both serum and islets compared to wildtype mice. Ultrastructural abnormalities in *Pdia1*-null ß cells include diminished insulin granule content, ER vesiculation and distention, mitochondrial swelling and nuclear condensation. Furthermore, *Pdia1* deletion increased accumulation of disulfide-linked high molecular weight proinsulin complexes and islet vulnerability to oxidative stress. These findings demonstrate that PDIA1 contributes to oxidative maturation of proinsulin in the ER to support insulin production and ß cell health.

DOI: https://doi.org/10.7554/eLife.44528.001

*For correspondence:
parvan@umich.edu (PA);
rkaufman@sbpdiscovery.org (RJK)

**Competing interests:** The authors declare that no competing interests exist.

## Introduction

Type two diabetes mellitus (T2D) is a complex disease caused by multiple genetic and environmental factors with an overarching problem of insufficient insulin to meet the level of insulin resistance (*Mokdad et al., 2001*; *Sladek et al., 2007*; *Narayan et al., 2007*; *Støy et al., 2007*; *Kaul and Ali, 2016*). Significant evidence supports the notion that pancreatic ß cell failure is fundamental in the etiology of T2D (*Alejandro et al., 2015*). Insulin resistance increases the burden placed on pancreatic ß cells to increase insulin synthesis and secretion that leads to associated defects including decreased ß cell number (*Clark et al., 1988*; *Butler et al., 2003*; *Yoon et al., 2003*), chronic ER stress and/or oxidative stress (*Robertson et al., 2007*; *Scheuner and Kaufman, 2008*; *Back and Kaufman, 2012*; *Han et al., 2015*), and a loss of ß cell identity (*Talchai et al., 2012*; *Swisa et al., 2017*).

Insulin biosynthesis is a complex and dynamically regulated process. Insulin production is dependent upon differentiation-specific gene expression, translation and translocation of preproinsulin into the ER, ER chaperone-facilitated protein folding and disulfide bond formation, vesicular transport of proinsulin from the ER through the Golgi compartments, and stimulus-coupled granule release (*Liu et al., 2018*). Protein sorting and assembly of proinsulin into nascent granules in the

*trans* Golgi network sets the framework for proteolytic processing of proinsulin and condensation of insulin with zinc to create mature secretory granules that are staged for secretion in response to stimuli (*Liu et al., 2014*). Well characterized bottlenecks in protein secretion occur at the stages of correct protein folding in the ER, anterograde trafficking through the secretory pathway and defective stimulus-coupled granule exocytosis. A more thorough understanding of insulin biogenesis should facilitate development of new and highly efficacious treatments for T2D that are based upon enhancing insulin output while preventing the loss of functional ß cells.

Proper oxidative protein folding by the formation of disulfide bonds in the ER is important for protein stability. Misfolded proteins in the ER can be retro-translocated to the cytosol and degraded via the ubiquitin proteasome system (*Wu and Rapoport, 2018*) and/or autophagy (*Loi et al., 2018*). Accumulation of misfolded proteins in the ER activates the unfolded protein response (UPR) through the ER stress transducers PERK, IRE1, and ATF6 to alleviate and adapt to the cellular stress. However, chronic stress from an inability to resolve protein misfolding can compromise cell health (*Wang and Kaufman, 2016*). The vitality of ER homeostasis for ß cell health is underscored by the development of diabetes in rodents and humans with defects that either cause ER protein misfolding or fail to respond when misfolding occurs (*Back and Kaufman, 2012*).

Proinsulin forms a native folded structure in the ER by disulfide bond formation comprised of two linkages between the A and B polypeptide chains (A7-B7 and A20-B19) and one in the A chain (A6-A11) (*Liu et al., 2018*; *Dai and Tang, 1996*; *Jia et al., 2003*; *Yan et al., 2003*). Mutations within proinsulin that impact disulfide bond formation cause neonatal diabetes in humans and *Akita* mice, serving as a model of proinsulin misfolding-induced ß cell failure (*Støy et al., 2007*; *Colombo et al., 2008*; *Riahi et al., 2018*). In general, disulfide bond formation within secretory proteins occurs during the early stages of protein folding as cysteine residues establish proximity to one another; however, enzymes can assist catalyzing this process (*Bulleid, 2012*). The specific complement of cellular redox machinery required for normal insulin output or to maintenance of insulin secretion under conditions of nutrient excess, obesity, or genetic predisposition to diabetes is undefined.

During disulfide bond formation, ER oxidoreductin 1 (ERO1) transfers oxidizing equivalents from $O_2$ to form disulfide bonds in a large family of ER oxidoreductases (*Hudson et al., 2015*); this process ultimately culminates in the transfer of electrons from sulfhydryls to molecular oxygen. This oxidoreductase family localized to the ER comprises over 17 members in mammals and each one interacts with specific substrates in a different manner (*Jessop et al., 2009*; *Braakman and Bulleid, 2011*). Among them, PDIA1, also known as prolyl 4-hydroxylase beta (*P4HB*), is the major oxidoreductase located in the ER lumen (*Ramming and Appenzeller-Herzog, 2012*; *Benham, 2012*).

Protein disulfide isomerase (PDI) can catalyze oxidation, reduction, and isomerization of disulfide bonds in the ER (*Freedman et al., 1994*). PDI has four domains (a, a', b, and b') with a linker (x) and acidic C-terminal region (c) where a KDEL ER retention signal resides (*Hatahet and Ruddock, 2007*). Similar to thioredoxin, the two catalytic domains (a, a') are separated by non-catalytic domains (b, b') to form a structure of a-b-b'-x-a'-c where each catalytic domain has an active site consisting of Cys-Gly-His-Cys residues (*Hatahet and Ruddock, 2007*). Thiol groups on cysteine residues in the active motif are responsible for the oxidoreductase activity, enabling PDI to catalyze both disulfide bond formation as well as disulfide bond isomerization for selective substrates (*Hudson et al., 2015*; *Hatahet and Ruddock, 2007*; *Freedman, 1995*; *Wang et al., 2015*). In addition, PDI also exhibits chaperone activity, independent of its disulfide isomerase-activity as it binds to misfolded proteins to prevent their aggregation (*Puig and Gilbert, 1994*; *Wilson et al., 1998*).

Although PDIA1 catalyzes both disulfide bond formation and isomerization of proinsulin in vitro (*Winter et al., 2011*), it remains unknown how PDIA1 influences proinsulin folding and insulin production in vivo. Here, we generated ß cell specific *Pdia1*-deleted mice and established that PDIA1 is required for optimal insulin production needed to maintain glucose homeostasis in the face of metabolic challenge. The results support the conclusion that therapeutics directed to promote native disulfide bond formation within proinsulin is an attractive strategy to prevent ß cell failure in T2D.

## Results

### Generation of ß cell-specific *Pdia1* deleted mice

As PDIA1 is highly expressed in islets (*Cras-Méneur et al., 2004*; *Ahmed and Bergsten, 2005*), we pursued analysis of ß cell-specific conditional *Pdia1*-null mice using tamoxifen (Tam)-regulated deletion of *floxed Pdia1* alleles (*Kim et al., 2013*) through rat insulin promoter driven Cre-recombinase (*RIP-Cre^ERT*) (*Figure 1A*) (*Dor et al., 2004*). Quantitative RT-PCR (qRT-PCR) demonstrated an ~75% decrease in *Pdia1* mRNA in isolated islets from the ß cell-specific *Pdia1*-knock out mice (*Pdia1 fl/fl*; *Cre^ERT* herein, KO, but genotypes are defined in the figures) with no effects on *Insulin 2*, *Pdia6* or *Pdia3* mRNAs (*Figure 1B*). Early studies demonstrated that mice with or without the *RIP-Cre^ERT* allele did not show significant differences in glucose homeostasis after a 14 wk HFD (*Figure 1—figure supplement 1A*). Therefore, we compared mice with two floxed alleles (*fl/fl*) and mice with one floxed and one wildtype (WT) allele (*fl/+*) with littermates that also harbor the *RIP-Cre^ERT* transgene, both before and/or after Tam injection. Western blotting of isolated islets from Tam-treated mice with the *RIP-Cre^ERT* allele demonstrated significantly reduced PDIA1 protein with increased expression of the UPR genes BiP, PDIA6 and GRP94 (*Figure 1C–D*), suggesting *Pdia1* deletion may cause ER stress in ß cells of the KO mice. Immunohistochemistry (IHC) of pancreas tissue sections with antibodies specific for proinsulin/insulin, glucagon and PDIA1 confirmed the absence of PDIA1 in a ß cell-specific manner in the KO mice (*Figure 2A–B, compare middle panels*). Interestingly, analysis of glucagon staining in pancreas sections detected very low PDIA1 expression in islet α cells (*Figure 2C, middle panels*), and we note that proglucagon contains no disulfide bonds.

Although the *RIP-Cre^ERT* allele was reported to be expressed in the hypothalamus (*Wicksteed et al., 2010*), Western blotting of PDIA1 in hypothalamic tissue did not detect reduced PDIA1 expression (*Figure 1—figure supplement 1B*). In addition, we previously reported that this *RIP-Cre^ERT* allele did not affect serum dopamine, which is synthesized in the arcuate nucleus of the hypothalamus (*Hahm et al., 2013*). Although we detected Cre positive staining in the hypothalamus, there was no difference in expression of growth hormone-releasing hormone (GHRH) in the KO mice compared to the control genotypes with the *RIP-Cre^ERT* allele.

Young male and female mice did not exhibit any defects in glucose homeostasis. However, on a regular chow diet, the challenge of aging in KO males caused statistically significant increase in glucose intolerance (*Figure 2D–E*).

### KO mice fed a high fat diet (HFD) become glucose intolerant with defective insulin production

To determine the role of ß cell PDIA1 in the face of metabolic stress, male mice were fed a 45% HFD. Genetic controls (*Pdia1 fl/fl* and *fl/+*) and KO mice fed HFD up to 32 wks showed no significant differences in body weight or weight gain (*Figure 3A*). However, fasting blood glucose (4 hr) in the HFD mice was significantly elevated in the KO versus genetic controls (*Figure 3B*). In addition, glucose intolerance was observed in HFD-fed KO mice as measured by the difference in the area under the glucose excursion curve (Delta-AUC) (*Figure 3C*). Serum insulin levels were decreased in HFD-fed KO mice, and a fasting-refeeding challenge revealed hypoinsulinemia with an increased proinsulin/insulin ratio (*Figure 3D*). *Pdia1* deletion did not affect insulin sensitivity measured by insulin tolerance tests (*Figure 3E*) and no significant difference was observed in the percent of ß cell area to total pancreas or ß cell area per islet in HFD-fed KO mice (*Figure 3—figure supplement 1*). These results indicate that in the setting of metabolic challenge, *Pdia1* is required for adequate insulin production to maintain systemic glucose homeostasis.

### *Pdia1* is not required for expression of ß cell-specific genes, antioxidant response genes or cell death genes

To reveal whether *Pdia1* deletion impacts gene expression to cause ß cell dysfunction, we isolated islets from male mice after 30 wks of HFD and analyzed mRNA levels by qRT-PCR. The results demonstrated no significant decrease in ß cell- and α cell- specific mRNAs, mRNAs encoding the insulin processing enzymes PC1/3, PC2, and CPE, or mRNAs encoding ß cell transcription factors PDX1 or MAFA (*Figure 3F*). Therefore, the reduced insulin content in KO serum and islets (*Figure 3D*) was not due to reduced ß cell-specific gene expression. There was also no significant change in

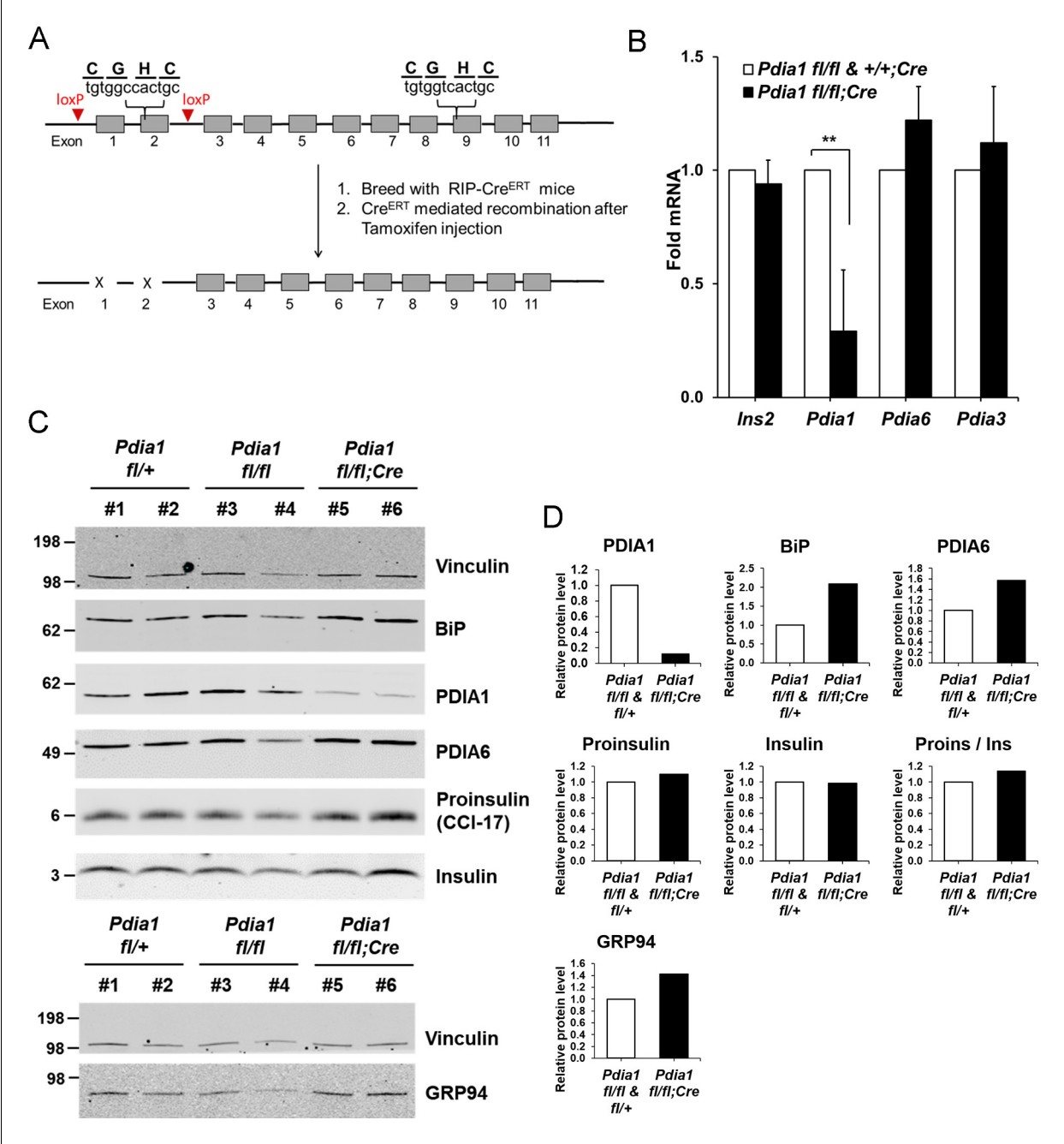

**Figure 1.** Conditional ß cell-specific *Pdia1* deleted mice were generated with Tamoxifen (Tam) induction. (**A**) Diagram depicts the generation of *Pdia1: RIP-Cre^ERT* mice. Mice with floxed *Pdia1* alleles (*Hahm et al., 2013*) were crossed with *RIP-Cre^ERT* transgenic mice (*Dor et al., 2004*) and progeny were injected IP with Tam to induce *Cre^ERT* function and *Pdia1* deletion. Control littermate mice with one or two floxed *Pdia1* alleles, but not harboring the *RIP-Cre^ERT* transgene, were injected in parallel with Tam. (**B–D**) *Pdia1* deletion is specific. (**B**) Total RNA was extracted from islets isolated from female mice at eight wks after Tam injection. mRNA levels were measured by qRT-PCR. Mean ± SEM, n = 3 for each group (p<0.01**). (**C**) Western blot illustrates expression of Vinculin, PDIA1, BiP, PDIA6, GRP94, Proinsulin, and Insulin in islets isolated from female mice at 14 wks after Tam injection. (**D**) Quantification of indicated proteins by Western blotting (from **C**) is shown. Each value was normalized to vinculin except for the proinsulin to insulin ratio. *Pdia1 fl/fl* (n = 2), *Pdia1 fl/+* (n = 2), *Pdia1 fl/fl;Cre* (n = 2).

DOI: https://doi.org/10.7554/eLife.44528.002

The following figure supplement is available for figure 1:

**Figure supplement 1.** The *RIP-Cre^ERT* allele does not impact the ß cell-specific *Pdia1* deletion phenotype.

DOI: https://doi.org/10.7554/eLife.44528.003

expression of other PDI family members and *Serca2b,* except for *Pdia4* (*Figure 3G*). In addition, UPR genes were not significantly elevated at this point in time in the *Pdia1* KO islets, other than *Grp94* (*Figure 3H*), which correlated with a slight increase in protein (*Figure 1C*). Lastly, there were no significant differences in expression of a panel of genes representing the antioxidant response and cell death (*Figure 3I*). These results show that ß cell-specific *Pdia1* deletion does not alter expression of ß cell specific genes, antioxidant response genes, or cell death genes.

## β cell *Pdia1* KO mice fed HFD exhibit ß cell failure with distinct morphological aberrations

The impact of *Pdia1* deletion on ß cell ultrastructure was analyzed by transmission electron microscopy (TEM) (*Figure 4*). TEM did not detect any significant morphological alterations in the cohorts of KO mice fed a regular diet for 10 wks after Tam injection (data not shown). Strikingly however, ß cells from HFD-fed KO mice showed significant abnormalities including ER vesiculation and distension, mitochondrial swelling, and nuclear condensation, that were not observed in genetic control mice (*Pdia1 fl/fl* and *fl/+*) (*Figure 4*). The red asterisks represent significantly distended ER, reflecting ER stress (*Figure 4D,H*, expanded). To determine whether *Pdia1* deletion also affects insulin granule content, we quantified the number of mature (dense dark core, yellow arrows in enlarged image) and immature (inner gray core, orange filled open arrowheads in enlarged image) granules and discovered that KO mice had fewer mature (~15%) and immature (~50%) granules compared to the control genotypes *fl/fl* and *fl/+* (*Figure 4I*). Analysis of the cross-sectioned mature granule (MG) vesicle area and dense core size demonstrated that the average cross-sectional area of MGs in KO mice was 20% larger than control genotypes (*Figure 4J*), however, there was no difference in MG dense core size between genotypes (*Figure 4J*), suggesting that insulin packaging efficiency within granules was decreased in *Pdia1* deleted-ß cells. This is consistent with decreased serum insulin levels in metabolically-challenged KO mice (*Figure 3D*). Taken together, the findings indicate that *Pdia1* is essential to maintain ß cell ultrastructure upon metabolic stress, indicative of suboptimal ß cell function.

## KO islets have an increased intracellular proinsulin/insulin ratio with accumulation of high molecular weight (HMW) proinsulin complexes

The impact of *Pdia1* deletion on proinsulin folding was investigated by analysis of proinsulin and insulin steady state levels in islets isolated from male mice after 30 wks of HFD using Western blotting under reducing and non-reducing conditions. The level of PDIA1 protein was decreased in islets isolated from KO mice, as expected (*Figure 5A–B*). Insulin levels were lower in KO islets compared to control genotypes while the proinsulin/insulin ratio was significantly elevated (*Figure 5A–B*). BiP/ *Hspa5* was induced with increased PDIA4 and PDIA6 in *Pdia1*-deleted versus *Pdia1*-sufficient islets (*Figure 5A–B*). The increased levels of BiP are consistent with the notion that the absence of *Pdia1* causes a mild ER stress, suggestive of protein misfolding, which is also consistent with ER distension (*Figure 4*).

The potential role of PDIA1 in disulfide maturation encouraged us to look for the presence of proinsulin disulfide-linked complexes by non-reducing SDS-PAGE (*Figure 5C*). Western blot analysis under non-reducing conditions using the CCI-17 antibody demonstrated the appearance of multiple proinsulin bands including a monomer at ~6 kDa (black arrowhead, *Figure 5C*) and a ladder of molecular masses that could correspond to proinsulin dimers up to pentamers (b, 14–49 kDa) as well as a cluster of HMW complexes (a, 49–198 kDa) in *Pdia1 fl/fl* and *KO* islets (*Figure 5C*). The nature of the disulfide-linked oligomeric proinsulin species was described elsewhere (*Arunagiri et al., 2019*). *Pdia1* deletion increased the amount of HMW complexes relative to the oligomeric forms and decreased the oligomers relative to monomeric proinsulin compared to the genetic controls without the *RIP-Cre$^{ERT}$* allele (*Figure 5C*, quantified in graph).

During the course of our studies we discovered that the epitope reactivity of the CCI-17 monoclonal antibody is very dependent the status of proinsulin sulfhydryls and disulfide bonds. Two major findings support this conclusion. First, when the non-reducing SDS-PAGE gel was treated with dithiothreitol (DTT) prior to transfer to nitrocellulose, we observed a great increase in antibody reactivity with monomeric proinsulin (*Figure 5C–D*). This suggests that opening the proinsulin molecule significantly enhances epitope exposure to the CCI-17 antibody. It is also important to note that

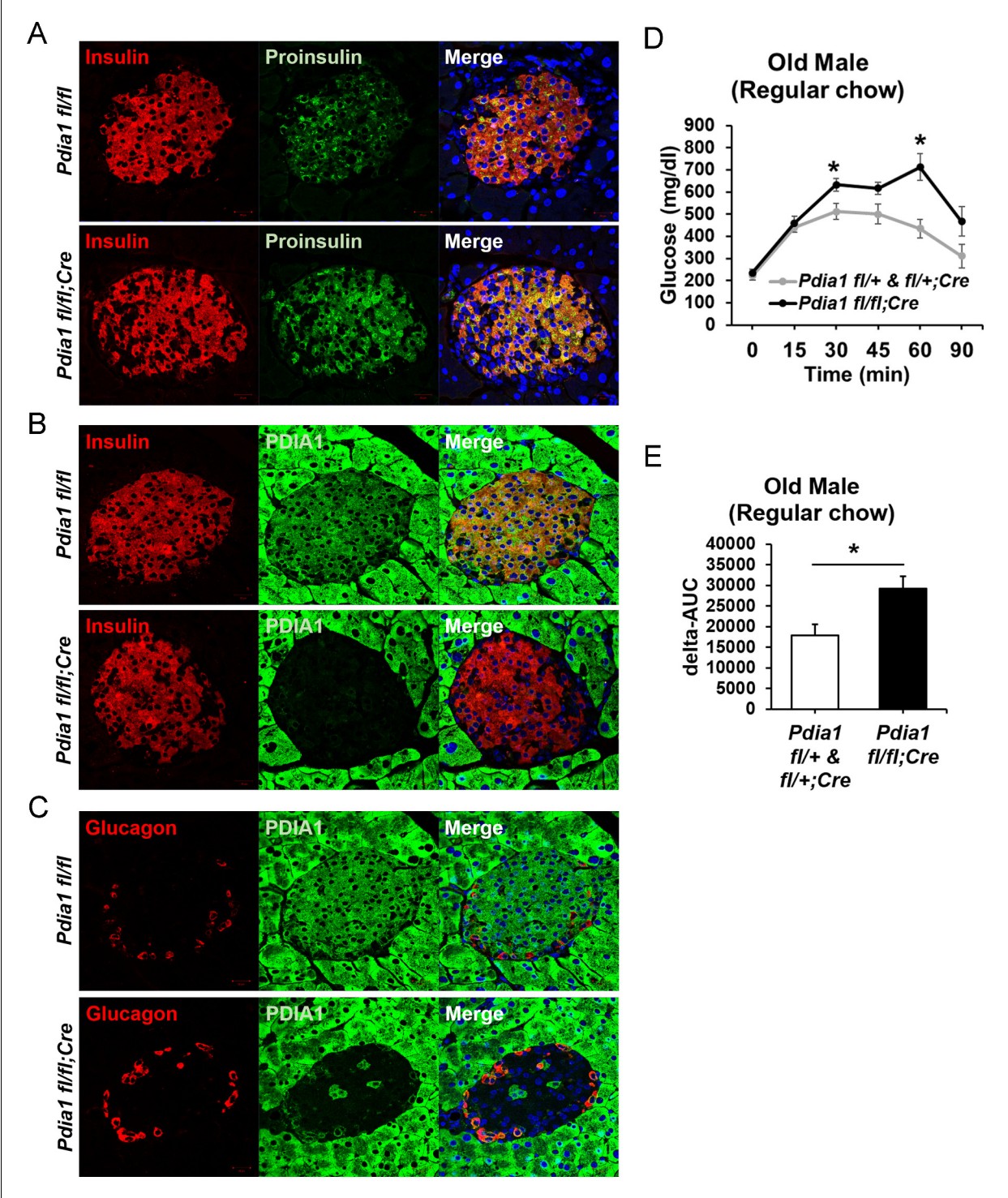

**Figure 2.** *Pdia1* is specifically and persistently deleted in murine ß cells. (A–C) Pancreas tissue sections were prepared from female mice at 49 wks after Tam injection and immuno-stained with anti- proinsulin, insulin, PDIA1, and glucagon antibodies. Images were merged with DAPI stain. Scale bar, 20 μm. (D) Old KO mice developed glucose intolerance compared to control genotypes measured by glucose tolerance testing (GTT) at nine wks after Tam injection. Male mice at 9 mon of age were injected with Tam and fed a regular chow. Mice were fasted (4 hr) prior to IP glucose injection (2 g/Kg body weight) and glucose levels were measured by tail bleeding at each time point (0; non-injected, 15, 30, 60, 90 min). control genotypes: *Pdia1 fl/+* (n = 2), *Pdia1 fl/+;Cre* (n = 5), KO; n = 5. (E) Area under the GTT curve (Δ-AUC) of (D) is indicated in graph.
DOI: https://doi.org/10.7554/eLife.44528.004

Western blotting analysis of the non-reducing SDS-PAGE gel (with or without subsequent DTT treatment of the gel prior to transfer) demonstrated readily detectable proinsulin disulfide-linked complexes in WT murine islets, nondiabetic human islets and in prediabetic *db/db* islets prior to onset of hyperglycemia (*Arunagiri et al., 2019*). The second finding is that treatment with N-ethylmaleimide (NEM) to alkylate free sulfhydryls slightly increased the HMW proinsulin-containing complexes and slightly reduced the disulfide-linked oligomers (*Figure 5E*). We suggest that PDIA1 may be required to reduce the HMW complexes and NEM treatment inactivates PDIA1, thereby stabilizing the HMW complexes. Similar results were obtained with a selective PDIA1 inhibitor (see below, *Figure 6—figure supplement 1*).

To gain further insight into the nature of the disulfide-linked oligomers, islets isolated from WT male mice fed a regular diet were treated in culture with increasing concentrations of dithiothreitol (DTT) to increase the ER reduction potential. This treatment produced increasing amounts of the proinsulin monomer and residual disulfide-linked dimer (*Figure 5F*), consistent with findings of *Arunagiri et al. (2019)*. In addition, female KO mice fed a regular diet for 14 wks after Tam also showed increased HMW complexes relative to the disulfide-linked proinsulin oligomers and reduced oligomers relative to monomeric proinsulin compared to the genetic controls (*Figure 5—figure supplement 1*).

## Oxidant treatment of *Pdia1* KO islets increases accumulation of HMW proinsulin complexes

Because ER stress is linked with oxidative stress (*Han et al., 2015*; *Malhotra et al., 2008*), we tested whether *Pdia1* deletion confers increased sensitivity to oxidants by treating islets (isolated after 30 wks of HFD) with the vitamin K analog menadione (*Criddle et al., 2006*; *Loor et al., 2010*), which can render cells susceptible to apoptosis by increasing cytosolic calcium (*Gerasimenko et al., 2002*) with nuclear condensation (*Wyllie et al., 1984*). Menadione treatment significantly increased ROS as observed by CellROX Deep Red stain (*Figure 6A*) and ß cells with *Pdia1* deletion showed greater ROS accumulation than those of genetic control *fl/+* islets (*Figure 6B*). Furthermore, both the average size and the histogram-analyzed nuclear size after Hoechst 33342 staining demonstrated that menadione promoted nuclear condensation (*Figure 6C*) and this effect appeared to be greater in KO islets compared to the genetic controls (*Figure 6D*). To uncover how *Pdia1* deletion may affect the sensitivity of proinsulin maturation to perturbation by oxidants, we performed Western blotting of islet lysates under reducing and non-reducing conditions. Consistently, in the absence of menadione treatment, *Pdia1* deletion increased the HMW proinsulin complexes (a, 49–198 kDa) compared to WT islets, and this effect was even greater in islets treated with menadione (*Figure 6E*). Importantly, DTT treatment of the gel prior to nitrocellulose transfer demonstrated no difference between untreated and NEM-treated islets (*Figure 6F*). Taken together, these results show that ß cells lacking *Pdia1* are more sensitive to oxidizing conditions that promote the formation of HMW proinsulin complexes.

## Proinsulin accumulation in the ER increases oligomeric and HMW complexes

Our studies suggest that PDIA1 is required to prevent formation of HMW proinsulin complexes. Our complementary study (*Arunagiri et al., 2019*) demonstrated that increased proinsulin accumulation predisposes to oligomer and HMW complex formation. To test the notion that increased proinsulin expression, as in T2D, may exacerbate abnormal disulfide formation, we tested the effect of brefeldin A (BFA), which promotes retrograde COP1 trafficking from the *cis*-Golgi to the ER to prevent export of secretory proteins to the Golgi, and thus increasing their concentration in the ER. BFA treatment significantly increased proinsulin complex formation, thus supporting the notion that the aberrant multimers/HMW complexes are a consequence of increased proinsulin content in the ER (*Figure 5—figure supplement 2*). Treatment with the translation elongation inhibitor cycloheximide (CHX), did not significantly affect the results, other than an expected reduced proinsulin content, indicating the presynthesized proinsulin is subject to multimer/HMW complex formation when its abundance in the ER is increased.

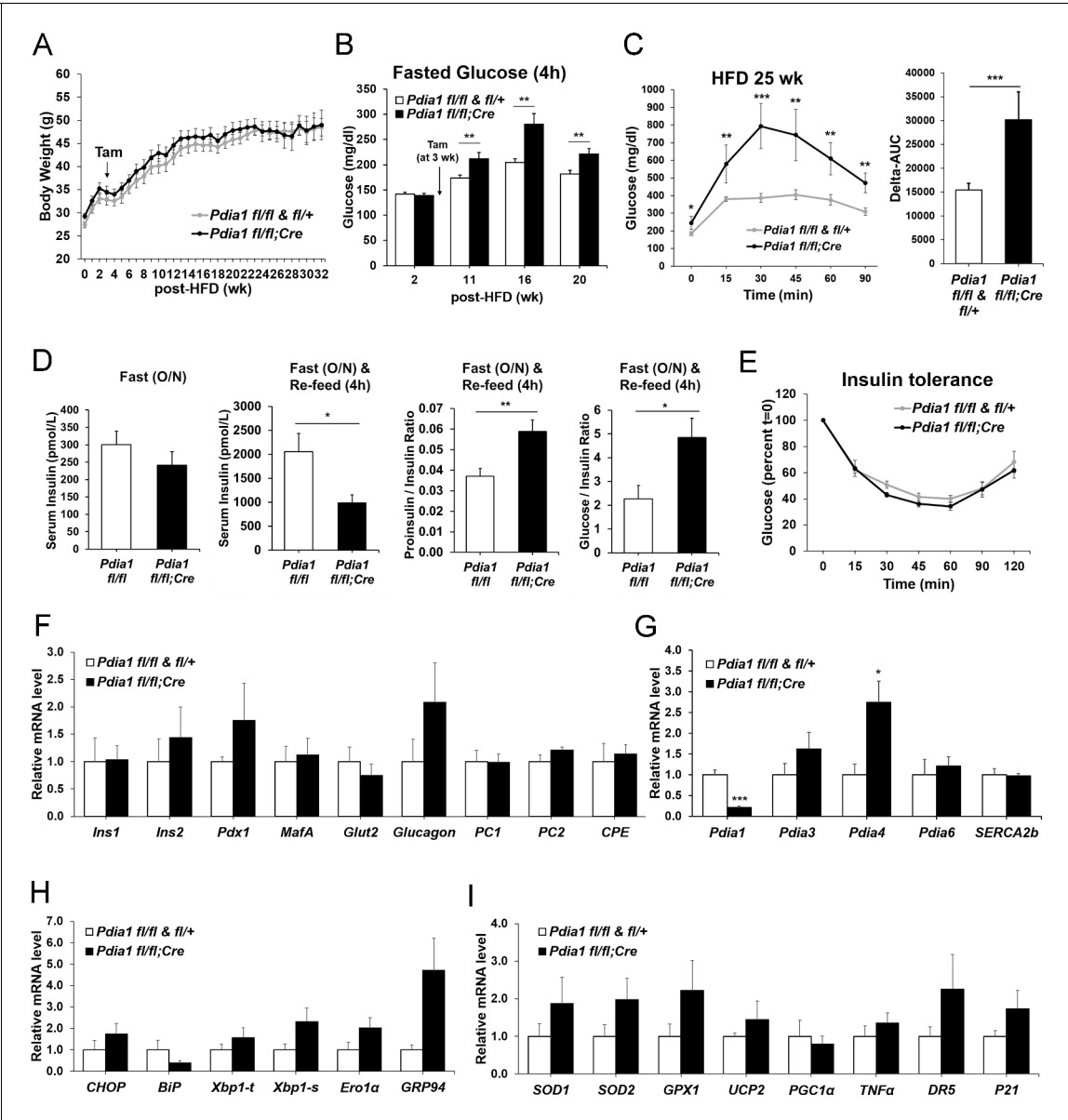

**Figure 3.** ß cell-specific *Pdia1* deleted male mice are glucose intolerant with defective insulin secretion when fed a 45% High Fat Diet (HFD). All mice were Tam injected at three wks after HFD was started. (**A**) No difference was observed in body weight (g) between control genotypes and KO mice. Mean ± SEM, controls; n = 17, KO; n = 12. (**B**) Fasting (4 hr) blood glucose levels were elevated in KO mice at 11, 16 and 20 wks after HFD. Glucose levels were measured by tail bleeding. Mean ± SEM, control genotypes; n = 17, KO; n = 12. (**C**) KO mice displayed higher blood glucose levels and area under the GTT curve (Δ-AUC) compared to control genotypes during glucose tolerance testing (GTT) after HFD for 25 wks. GTT were performed at multiple time points after HFD in two independent cohorts and representative results are shown. Mice were fasted (4 hr) prior to IP glucose injection (1 g/Kg body weight) and glucose levels were measured by tail bleeding at each time point (0; non-injected, 15, 30, 60, 90 min). control genotypes; n = 17, KO; n = 6. (**D**) KO mice exhibited decreased serum insulin levels and an increased serum proinsulin/insulin ratio compared to control genotypes. Insulin and proinsulin ELISAs were performed with the serum obtained from mice after fasting (overnight) and re-feeding (4 r) after HFD for 17 wks. control genotypes; n = 8, KO; n = 12. (**E**) No difference was observed in insulin tolerance tests performed after HFD for 20 wks. Mice were fasted for 4 hr before IP injection of insulin (1.5units/Kg). Glucose levels were measured by tail bleeding at each time point (0; non-injection, 15, 30, 60, 90, 120 min). control genotypes; n = 17, KO; n = 12. (**F–I**) Total RNA was extracted from islets isolated from mice after HFD for 30 wks. mRNA expression was measured by qRT-PCR. control genotypes; n = 3, KO; n = 3 mice. (**F**). ß cell-, α cell- and insulin processing genes. (**G**) PDI family and SERCA genes. (**H**) UPR genes. (**I**) Antioxidant response- and cell death- related genes. All data are shown as Mean ± SEM. p<0.05*, p<0.01**, p<0.001***.

DOI: https://doi.org/10.7554/eLife.44528.005

The following source data and figure supplement are available for figure 3:

*Figure 3 continued on next page*

*Figure 3 continued*

**Source data 1.** Primer sequences used for qRT-PCR.
DOI: https://doi.org/10.7554/eLife.44528.007
**Figure supplement 1.** ß cell area relative to pancreas area and β cell number relative to islet area were not changed in ß cell-specific *Pdia1* deleted male mice after 34 wks of HFD.
DOI: https://doi.org/10.7554/eLife.44528.006

## PDIA1 is a reductase that facilitates proper proinsulin folding

To study the impact of PDIA1 in the absence of C peptide processing in a more manipulatable system, we analyzed human proinsulin expression delivered by adenovirus to murine embryonic fibroblasts (MEFs) that were co-infected with WT PDIA1 or PDIA1 with 4 Cys to Ser mutations in the two vicinal catalytic PDI sites (*Wang et al., 2012*). Infection with Ad-hProins produced significant amounts of reduced and oxidized proinsulin upon analysis by non-reducing SDS-PAGE with DTT treatment prior to transfer to nitrocellulose (*Figure 5—figure supplement 3*). Treatment of cells with increasing concentrations of DTT increased the amount of reduced proinsulin (lanes 3,4), as expected. Significantly, forced expression of PDIA1, but not catalytically inactive PDIA1, increased the amount of reduced proinsulin (lanes 5, 6, 8, 10 and 12). Therefore, the results support the hypothesis that PDIA1 acts as a reductase to prevent aberrant proinsulin oxidation, consistent with previous studies in vitro (*Rajpal et al., 2012*).

## Pharmacological inhibition of PDIA1 recapitulates effects of *Pdia1* gene deletion

Recently, *Cole et al. (2018)* described a selective PDIA1 inhibitor that covalently interacts with the A catalytic motif in PDIA1. Therefore, we tested the effect of this inhibitor in murine WT islets. We also studied the impact of oxidant menadione treatment. The results show that the KSC-34 inhibitor alone slightly increased HMW complexes. However, when combined with menadione, KSC-34 treatment recapitulated the effect of *Pdia1* deletion upon menadione treatment (*Figure 6—figure supplement 1*). Importantly, the findings demonstrate that either a PDIA1 chemical inhibitor or gene deletion exhibit similar effects on the generation of HMW complexes, especially upon oxidant treatment.

## Discussion

PDIA1 is the major ER oxidoreductase in the majority of mammalian cells, including ß cells. Although numerous in vitro studies demonstrated that PDI actively engages proinsulin to catalyze disulfide bond formation (*Winter et al., 2011*; *Rajpal et al., 2012*; *Winter et al., 2002*; *Wright et al., 2013*), there is little information regarding the significance of PDI action in vivo. Here, using ß cell specific *Pdia1* deletion we show that PDIA1 is increasingly important for insulin production in the face of either age or metabolic stress imposed by a HFD. Specifically, when compromised by HFD feeding, mice with ß cell-specific *Pdia1* deletion displayed exaggerated glucose intolerance with significant ß cell abnormalities including diminished islet and serum insulin accompanied by an increased proinsulin/insulin ratio in islets and serum (*Figure 3*), with diminished insulin packaging and storage in secretory granules and a reduced number of insulin secretory granules (*Figure 4*). In addition, ß cell *Pdia1* deletion caused abnormal ultrastructural changes including ER distension and vesiculation, mitochondrial swelling, and nuclear condensation (*Figure 4*). *Pdia1*-deleted islets were also sensitive to oxidant challenge (*Figure 6*), which is significant because PDIA1 is a highly abundant ER protein (~mM concentration [*Freedman et al., 1994*]) that is primarily in a reduced form (*Hudson et al., 2015*) and although it cycles, it may significantly contribute to redox homeostasis in the ER.

PDIA1 might assist proinsulin folding by facilitating proper intramolecular disulfide bond formation, yet to date, there is no direct evidence to support this notion (*Rajpal et al., 2012*). Alternatively, PDIA1 may reduce improper proinsulin disulfide bonds as demonstrated during infection with pathogens that require reduction for retro-translocation from the ER to the cytosol (*Tsai et al., 2001*; *Inoue et al., 2015*) and from evidence that supports a role for PDI as a reductase important for degradation of mutant *Akita* proinsulin (*He et al., 2015*) and mutant thyroglobulin (*Forster et al., 2006*).

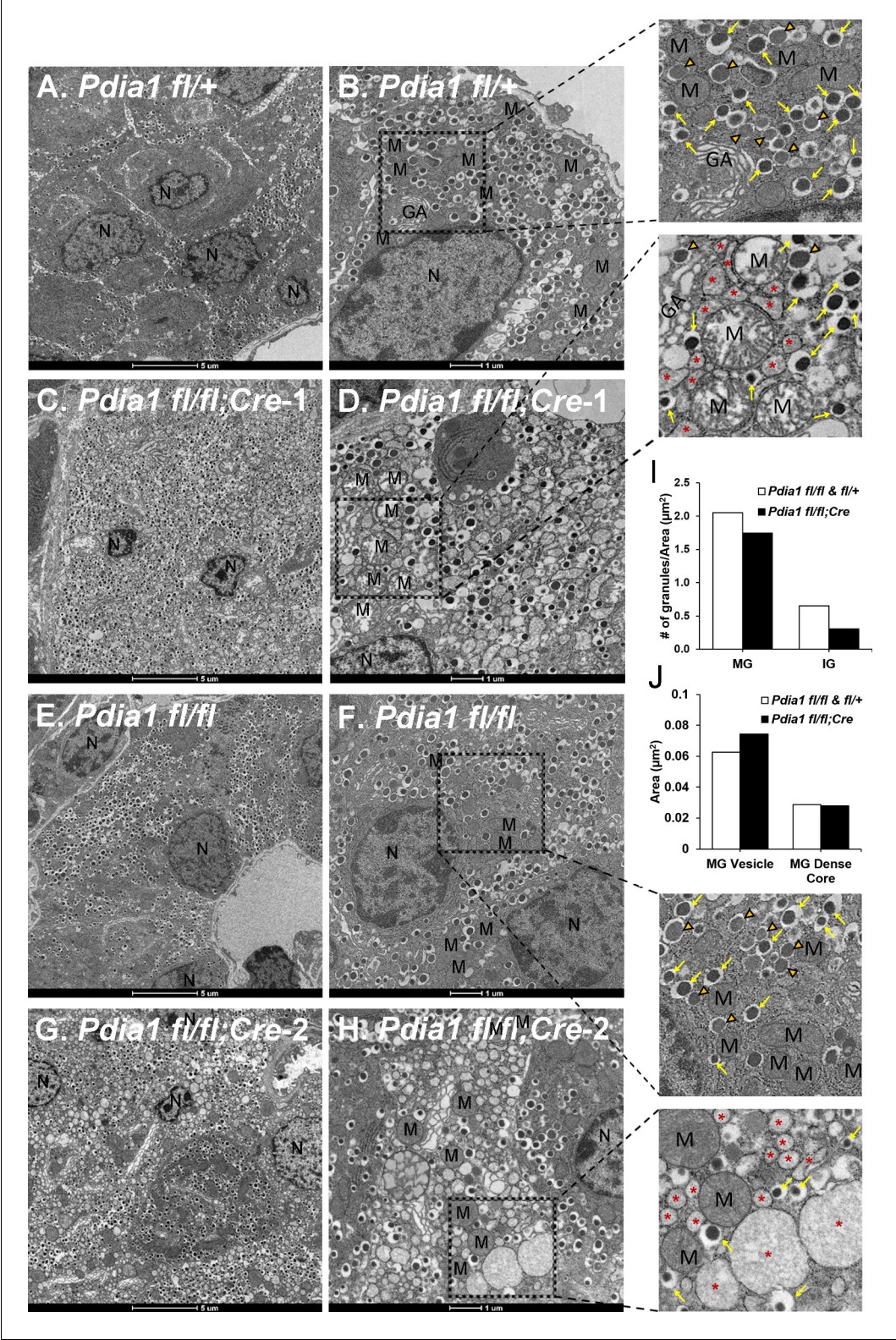

**Figure 4.** *Pdia1* deletion induces morphological abnormalities including decreased insulin granule numbers, ER vesiculation and distention, mitochondrial swelling and nuclear condensation in ß cells. (A–H). Transmission electron microscopy was performed on pancreata obtained from genetic controls (**A, B, E, F**) and KO (**C, D, G, H**) male mice after 40 wks of HFD. Images were obtained at 1900X (**A, C, E, G**) or 4800X (**B, D, F, H**) magnification. Scale bar represents 5 μm or 1 μm as indicated. Marked area is two times enlarged on the right side: N, nucleus; M, mitochondria; *,

*Figure 4 continued on next page*

*Figure 4 continued*

distended endoplasmic reticulum; GA, Golgi apparatus. Yellow arrows, mature granules. Orange filled open arrowheads, immature granules. (**I**) *Pdia1* KO mice had reduced numbers of mature and immature granules. For each genotype 80–110 images were quantified. genetic controls; n = 3, KO; n = 2. Mean value is indicated. (**J**) Mature granule sizes were larger in *Pdia1* KO mice compared to genetic controls without differences in mature granule dense core size. For each genotype, 40 images were quantified. genetic controls; n = 3, KO; n = 2.

DOI: https://doi.org/10.7554/eLife.44528.008

Proinsulin disulfide maturation in the ER is absolutely required for proinsulin export to the Golgi complex for delivery to immature granules (*Haataja et al., 2016*). The identification of mutations in the *INS* gene coding sequence increased our understanding how proinsulin misfolding contributes to the development of ß cell failure and diabetes (*Liu et al., 2010*). If PDIA1 assists in disulfide isomerization to facilitate correct disulfide bonding in proinsulin, its absence could increase aberrant disulfide-linked proinsulin complexes, which we observed in *Pdia1*-deleted islets (*Figure 5*). Although the role of PDIA1 in vivo is complicated by the presence of other ER oxidoreductases and glutathione, based on the greater accumulation of HMW disulfide-linked proinsulin complexes in *Pdia1*-null islets, the data strongly suggest that PDIA1 participates in the resolution/dissolution of these inappropriate disulfide-linked complexes (*Figure 7*). This could include both PDI chaperone function as well as oxidoreductase function. It is important to note that healthy murine WT islets also exhibit a lower level of these HMW complexes in addition to smaller oligomeric disulfide-linked proinsulin species (*Arunagiri et al., 2019*). Future studies will characterize the fate of these complexes, and how PDIA1 plays a role in limiting their accumulation and their potential contributions to relative insulin deficiency and ß cell failure, which are phenotypes known to be associated with T2D. Importantly, the increased BiP expression in the islets bearing *Pdia1*-deleted ß cells suggests that increased accumulation of inappropriate disulfide-linked proinsulin complexes may induce ER stress, supporting a link between aberrant disulfide-linked proinsulin complexes and a compromise in ß cell health and function.

Based on our results, we conclude that PDIA1 optimizes proinsulin maturation needed for insulin biosynthesis and for the maintenance of normoglycemia under conditions of metabolic stress in vivo. Our findings should open a new field of investigation to elucidate how the PDI family members impact oxidative protein folding and redox homeostasis in the ER of pancreatic ß cells, as well as provide a fundamental basis to understand how protein folding is essential to protect ß cells from collapse, contributing to the onset of T2D.

# Materials and methods

**Key resources table**

| Reagent type (species) or resource | Designation | Source or reference | Identifiers | Additional information |
|---|---|---|---|---|
| Gene (*M. musculus*) | *Pdia1/P4hb* | NA | MGI:97464 | |
| Strain, strain background (*M. musculus*) | *Pdia1 fl/fl* | *Hahm et al., 2013* | | |
| Strain, strain background (*M. musculus*) | *RIP-Cre*[ERT] | *Dor et al., 2004* | | |

*Continued on next page*

*Continued*

| Reagent type (species) or resource | Designation | Source or reference | Identifiers | Additional information |
|---|---|---|---|---|
| Strain, strain background (*M. musculus*) | *Pdia1 fl/fl;RIP-Cre^ERT* | this paper | | C57BL/6 mice with Pdia1 floxed alleles were obtained from Dr. J Cho (Univ. of Illinois-Chicago) and crossed with Rat Insulin Promoter (RIP-CreERT) transgenic mice. Congenic Pdia1 gene floxed littermates with or without the CreERT transgene were used for in vivo experiments. |
| Cell line (*M. musculus*) | WT MEF | this paper | WT MEF | Freshly prepared primary WT MEFs prior to passage #5 were used for experiments. |
| Adenovirus | human PDI | *Wang et al., 2012* | | |
| Adenovirus | human PDImut | *Wang et al., 2012* | | |
| Adenovirus | human Proins | this paper | | Human proinsulin was cloned into the pAd Easy system. |
| Antibody | Mouse monoclonal anti-Vinculin | Proteintech | Cat. #: 66305–1-Ig | WB (1:2000) |
| Antibody | Rabbit monoclonal anti-GRP94 | Cell signaling Technology | Cat. #: 20292P | WB (1:1000) |
| Antibody | Mouse monoclonal anti-BiP | BD Biosciences | Cat. #: 610979 | WB (1:1000) |
| Antibody | Rabbit polyclonal anti-PDIA1 | Proteintech | Cat. #: 11245–1-AP | WB (1:1000), IHC (1:200) |
| Antibody | Rabbit polyclonal anti-PDIA4 | Proteintech | Cat. #: 14712–1-AP | WB (1:1000) |
| Antibody | Rabbit polyclonal anti-PDIA6 | Proteintech | Cat. #: 18233–1-AP | WB (1:1000) |
| Antibody | Mouse monoclonal anti-Proinsulin | HyTest Ltd. | Cat. #: 2PR8 (Mabs: CCI-17) | WB (1:10000), IHC (1:200) |
| Antibody | Guinea pig polyclonal anti-Insulin | this paper | | Produced in house WB (1:2000), IHC (1:200) |
| Antibody | Mouse monoclonal anti-Glucagon | Abcam | Cat. #: K79bB10 | IHC (1:200) |
| Antibody | IRDye 800CW Goat anti-Mouse IgG (H + L) | Li-Cor | P/N: 926–32210 | WB (1:5000) |
| Antibody | IRDye 680RD Goat anti-Mouse IgG (H + L) | Li-Cor | P/N: 926–68070 | WB (1:5000) |

*Continued on next page*

*Continued*

| Reagent type (species) or resource | Designation | Source or reference | Identifiers | Additional information |
|---|---|---|---|---|
| Antibody | IRDye 800CW Goat anti-Rabbit IgG (H + L) | Li-Cor | P/N: 926–32211 | WB (1:5000) |
| Antibody | IRDye 680RD Goat anti-Rabbit IgG (H + L) | Li-Cor | P/N: 926–68071 | WB (1:5000) |
| Antibody | IRDye 800CW Donkey anti-Guinea Pig IgG (H + L) | Li-Cor | P/N: 926–32411 | WB (1:5000) |
| Antibody | IRDye 680RD Donkey anti-Guinea Pig IgG (H + L) | Li-Cor | P/N: 926–68077 | WB (1:5000) |
| Antibody | Alexa Fluor 488 goat anti-rabbit IgG | Invitrogen | Cat. #: A-11008 | IHC (1:500) |
| Antibody | Alexa Fluor 488 goat anti-mouse IgG | Invitrogen | Cat. #: A-11001 | IHC (1:500) |
| Antibody | Alexa Fluor 594 goat anti-mouse IgG | Invitrogen | Cat. #: A-11005 | IHC (1:500) |
| Antibody | Alexa Fluor 594 goat anti-guinea pig IgG | Invitrogen | Cat. #: A-11076 | IHC (1:500) |
| Commercial assay or kit | Mercodia Rat/Mouse Proinsulin ELISA | Mercodia | 10-1232-01 | |
| Commercial assay or kit | Mercodia Mouse Insulin ELISA | Mercodia | 10-1247-01 | |
| Commercial assay or kit | CellROX Deep Red reagent | Molecular Probes | C10422 | |
| Chemical compound, drug | DTT | Roche diagnostics | Product #.3117006001 | |
| Chemical compound, drug | Menadione | AdipoGen Life Science | AG-CR1-3631-G001 | |
| Chemical compound, drug | Cycloheximide | Sigma-Aldrich | C7698 | |
| Chemical compound, drug | Brefeldin A | Cell signaling Technology | Cat. #: 9972 | |
| Chemical compound, drug | KSC-34 | *Cole et al., 2018* | | Obtained from Dr. RL Wiseman (The Scripps Research Institute, La Jolla, CA) |

## Mice

C57BL/6 mice with *Pdia1* floxed alleles were obtained from Dr. J Cho (Univ. of Illinois-Chicago) (*Hahm et al., 2013*) and crossed with Rat Insulin Promoter (*RIP-Cre^ERT*) transgenic mice (*Dor et al., 2004*). Congenic *Pdia1* gene floxed littermates with or without the *Cre^ERT* transgene were used for in vivo experiments. *Pdia1* deletion was performed by IP injection of the estrogen receptor antagonist Tamoxifen (Tam) (4 mg/mouse) three times a week. Male mice were pair-housed for the high fat diet (HFD) study. All procedures were performed by protocols and guidelines reviewed and approved by the Institutional Animal Care and Use Committee (IACUC) at the SBP Medical Discovery Institute (AUF #. 17–066).

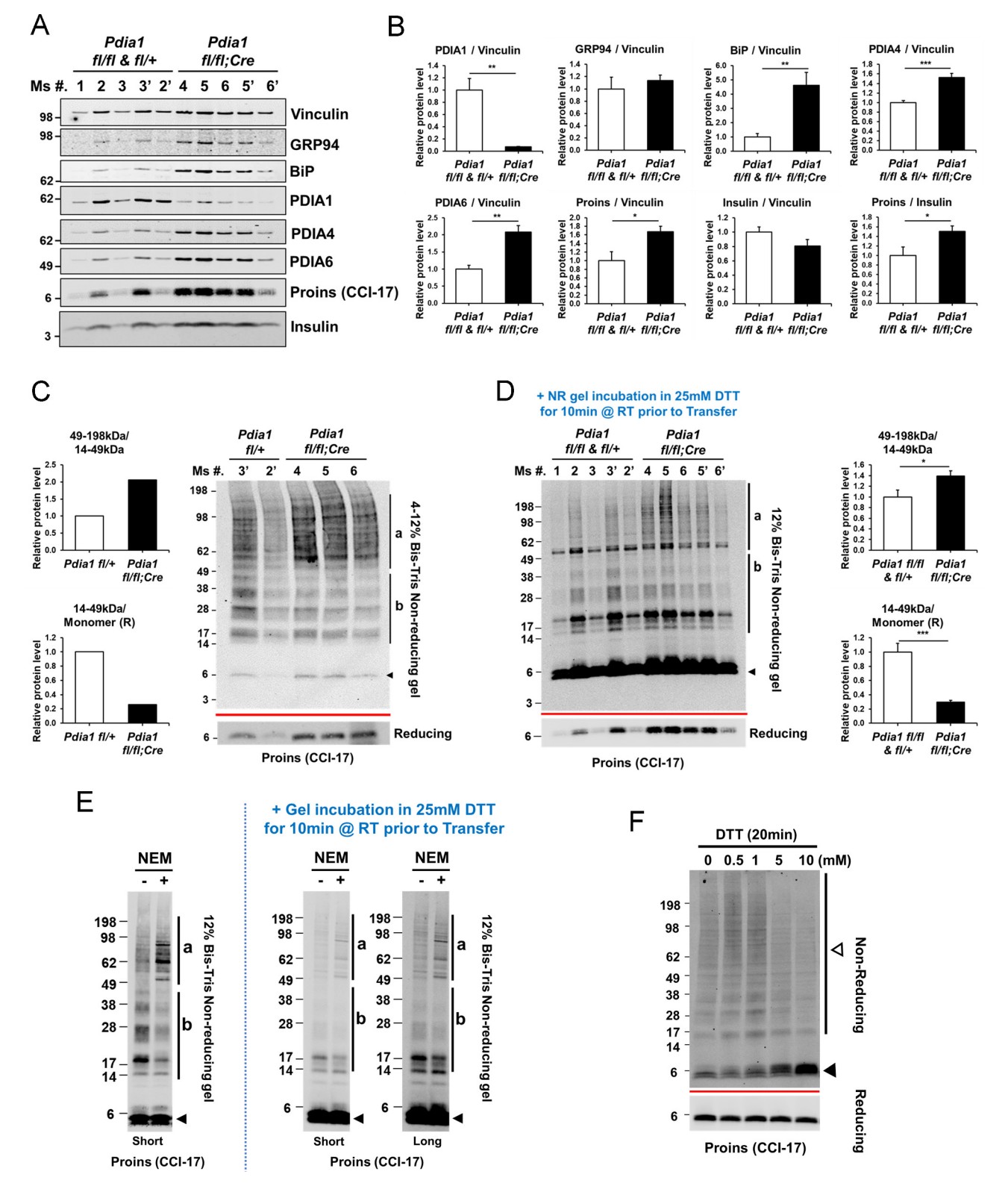

**Figure 5.** *Pdia1* deletion in HFD fed mice increases islet steady state proinsulin to insulin ratio with accumulation of high molecular weight (HMW) proinsulin complexes. (A, C) Western blotting was performed for murine islets isolated after HFD for 30 wks. After overnight recovery, islets were lysed and analyzed under reducing (A) or non-reducing (C, D) conditions by SDS-PAGE and Western blotting. Image exposed for a different time for reduced proinsulin (CCI-17) in (A) was used to represent the total proinsulin levels in (D). Six independent mice were analyzed and technical duplicates are

*Figure 5 continued on next page*

*Figure 5 continued*

indicated as' on the top of gel. (B). Quantification of indicated proteins was performed under reducing conditions (A). Each value was normalized to vinculin. Proinsulin/insulin ratios were calculated based on the quantification of proinsulin and insulin species under reducing conditions (A). (C) Five samples from A were analyzed under non-reducing conditions on a 4–12% Bis-Tris SDS gel. The proinsulin blot under reducing conditions is under the red line. Left side. Quantification of HMW proinsulin complexes under non-reducing conditions is indicated. (C), left side, upper) The ratio of 49–198 kDa (a) to 14–49 kDa (b) proinsulin complexes is shown. (C), left side, lower) The ratio of 14–49 kDa (b) proinsulin complexes to proinsulin monomer under reducing conditions is shown. Mean value is indicated in graph. (D) The ten samples in A were analyzed under non-reducing conditions after the gel was incubated in 25 mM DTT for 10 min at RT prior to transfer. To control for variable transfer from a gradient gel, we used a fixed percentage gel (12% Bis-Tris SDS). Right side. Quantification is shown for HMW proinsulin complexes under non-reducing conditions. (D), right side, upper) The ratio of 49–198 kDa (a) to 14–49 kDa (b) proinsulin complexes is shown. (D), right side, lower) The ratio of 14–49 kDa (b) proinsulin complexes to proinsulin monomer under reducing conditions is shown. (A–D) genetic controls; n = 3, KO; n = 3 mice. Mean ± SEM, p<0.05*, p<0.01**, p<0.001***. (E) WT murine islets were treated with or without NEM and lysates were analyzed under non-reducing conditions. Equal numbers of islets were divided into two tubes. Left side islets were rinsed with cold-PBS and lysed on ice. Right side islets were rinsed with cold-PBS containing 20 mM NEM and lysed in lysis buffer containing 2 mM NEM. Samples were prepared alongside and lysates were boiled for 5 min. Equal amounts of lysates were loaded and analyzed with or without gel incubation in 25 mM DTT for 10 min at RT prior to transfer. Two different exposure time images (short, long) are shown after DTT incubation. (F) WT murine islets were treated with increasing concentrations of DTT for 20 min in culture at room temperature and then analyzed by non-reducing and reducing SDS-PAGE and Western blotting with proinsulin antibody (CCI-17). The range of oligomeric proinsulin species are identified by an open arrowhead and monomeric proinsulin is indicated by black arrowhead.

DOI: https://doi.org/10.7554/eLife.44528.009

The following figure supplements are available for figure 5:

**Figure supplement 1.** *Pdia1* deletion increases accumulation of HMW proinsulin complexes under regular diet.
DOI: https://doi.org/10.7554/eLife.44528.010
**Figure supplement 2.** Inhibition of ER to Golgi trafficking increases proinsulin disulfide linked HMW complex formation.
DOI: https://doi.org/10.7554/eLife.44528.011
**Figure supplement 3.** PDIA1 overexpression reduces proinsulin.
DOI: https://doi.org/10.7554/eLife.44528.012

## Generation and culture of primary mouse embryonic fibroblast (MEF)

WT C57BL/6 d14 embryos were isolated under sterile conditions and placed in 100 mm cell culture dish containing PBS. Placental and other maternal tissues were removed and the embryos were washed three times with PBS. Heads and visceral organs were removed. The heads were used for genotyping. Embryos were finely minced with a sterile razor blade and treated with 1 ml of 0.25% trypsin/EDTA (Corning) for 40 min at 37°C. DMEM medium (Corning) supplemented with 10% FBS, 1% penicillin/streptomycin, 100 µg/ml primocin and 1 mM sodium pyruvate was added to quench trypsin activity. Tissue homogenate was subsequently disaggregated by repeated pipetting, and further centrifuged to collect the fibroblast cell pallet. Cells were plated in 100 mm culture dish in DMEM medium at 37°C in a cell culture incubator under 5% $CO_2$. To prevent mycoplasma, bacterial, and fungal contamination, primary MEFs were cultured in medium containing 100 µg/ml primocin (InvivoGen) and used for experiments prior to passage #5.

## Glucose and insulin tolerance tests

Glucose tolerance tests were performed by IP injection of glucose (1 g/Kg body weight) into mice after fasting for 4 hr. For insulin tolerance tests, 1.5units/Kg of insulin was injected IP into mice after a 4 hr fast. Blood glucose levels were measured by tail bleeding at each time point indicated.

## Measurement of serum proinsulin and insulin

Mice were fasted O/N and re-fed for 4 hr. Blood was collected by retro-orbital bleeding and serum was prepared by centrifugation. Serum proinsulin and insulin levels were measured by ELISA (Mercodia, 10-1232-01, 10-1247-01) according to the manufacturer's protocol.

## Islet isolation

Islets were isolated by collagenase P (Roche) perfusion as described (*Sutton et al., 1986*) following by histopaque-1077 (Sigma-Aldrich, Inc St. Louis) gradient purification. Islets were handpicked and studied directly or after overnight culture in RPMI 1640 medium (Corning 10–040-CV) supplemented

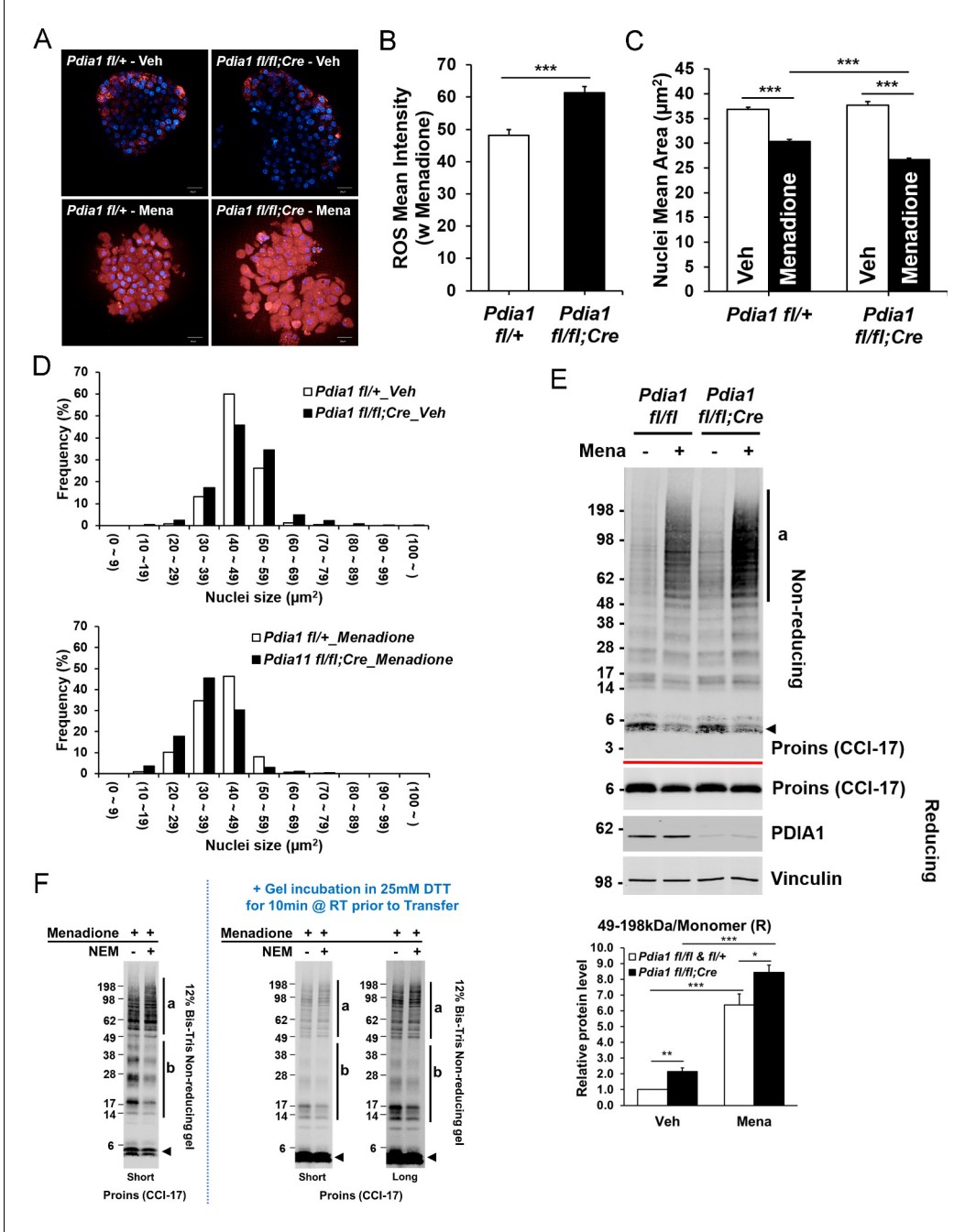

**Figure 6.** *Pdia1* deletion increases sensitivity to menadione oxidant: Increased ROS, nuclear condensation, and HMW proinsulin complexes were observed in menadione-treated KO islets. **A**. Islets isolated from mice after 30 wks HFD were treated with or without Menadione (10 μM, 3 hr) and co-stained with CellROX Deep Red (red) and Hoechst 33342 (blue). Live islet images were obtained by an Opera Phenix high content screening system (63X objective lens) and seven z-stack images (1 μm interval) were combined. Scale bar, 20 μm. genetic controls; n = 3, KO; n = 3. (**B**) Quantification of ROS mean intensity is shown. CellROX Deep Red mean intensity (divided by area) was measured by image J software. Mean ± SEM, p<0.001***. (**C**). Quantification of nuclear mean area (μm$^2$) measured in Hoechst 33342 stained images by ImageJ software is shown. Mean ± SEM, p<0.001***. (**D**). Histogram analysis of nuclear sizes is shown. Percent frequencies are indicated in the graph. (**E**) Western blot of islets isolated from mice after 37 wks of HFD by SDS-PAGE under reducing and non-reducing conditions is shown. After overnight recovery, islets were treated with menadione (100 μM) for 1 hr. Islet preparations from five independent control and KO mice were performed and representative images are shown. Quantification of the ratio of HMW proinsulin complexes (a) to monomeric proinsulin under reducing conditions is shown in graph (lower). Mean ± SEM, p<0.05*, p<0.01**, p<0.001***. Controls; n = 5, KO; n = 5 mice. (**F**) WT murine islets were treated with Menadione (100 μM) for 1 hr, treated with or without NEM as in *Figure 5E*, and lysates were prepared and analyzed under non-reducing conditions.

DOI: https://doi.org/10.7554/eLife.44528.013

*Figure 6 continued on next page*

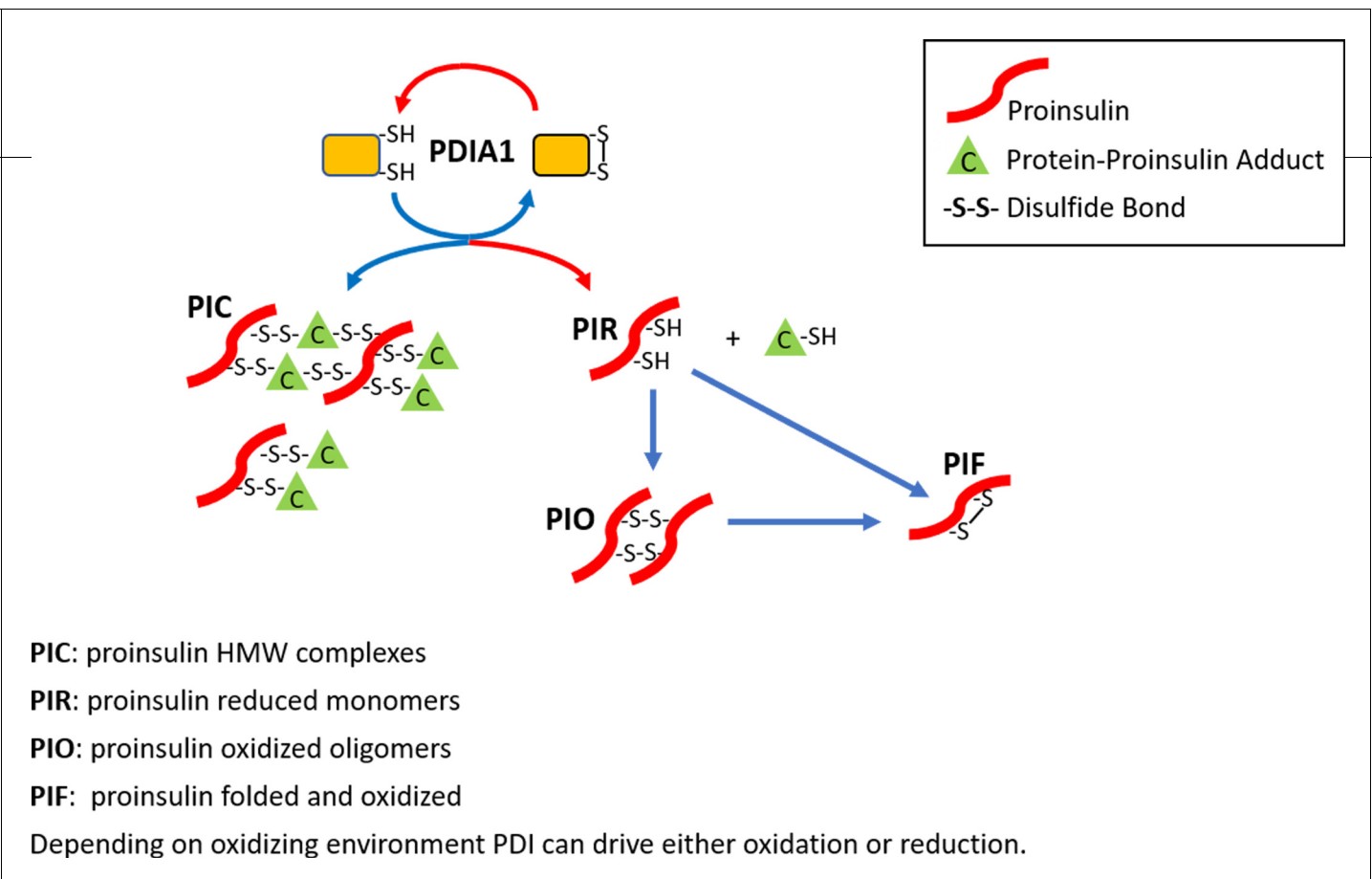

**Figure 7.** The role of PDIA1 in proinsulin disulfide bond formation. Here, we show that PDIA1 is not required but increases the efficiency of proinsulin maturation, possibly by reducing HMW proinsulin complexes. In the absence of PDIA1, disulfide bond formation, reduction and/or isomerization in proinsulin are inadequate during increased demand so HMW proinsulin complexes accumulate in the ER upon metabolic pressure, such as a HFD. The figure depicts the formation of proinsulin HMW complexes with cellular proteins and the role of PDIA1 in interconversions to reduced proinsulin with subsequent oxidation to dimer/oligomeric proinsulin and oxidized folded proinsulin.

DOI: https://doi.org/10.7554/eLife.44528.015

with 10% FBS, 1% penicillin/streptomycin, 100 µg/ml primocin, 10 mM Hepes, and 1 mM sodium pyruvate.

## Islet RNA isolation and qRT-PCR

Total RNAs were extracted from isolated islets by RiboZol Extraction reagent (VWR Life Science). cDNA was synthesized by iScript cDNA Synthesis kit (Bio-Rad Laboratories, Inc). The relative mRNA levels were measured by qRT-PCR with iTaq Universal SYBR green Supermix (Bio-Rad Laboratories, Inc). All primers are listed in *Figure 3—source data 1*.

## Islet Western blotting

Isolated islets were lysed in RIPA buffer (10 mM Tris pH 7.4, 150 mM NaCl, 0.1% SDS, 1% NP-40, 2 mM EDTA) with protease and phosphatase inhibitors (Fisher Scientific) on ice for 10 min and lysates were collected after centrifugation at 4℃ for 10 min at 12000 g. Samples were prepared in Laemmli sample buffer without (non-reducing) or with (reducing) 5% β-mercaptoethanol. After boiling for 5 min, samples were analyzed by SDS-PAGE (4–12% Bis-Tris gel, Bio-Rad Laboratories, Inc) and transferred to nitrocellulose membranes (Bio-Rad Laboratories, Inc). For DTT incubation prior to transfer, non-reduced samples were electrophoresed on a 12% Bis-Tris SDS gel and then incubated in 25 mM DTT for 10 min at RT. Primary antibodies were as follows: α-vinculin (Proteintech, 66305–1-Ig), α-

GRP94 (Cell Signaling, 20292P), α-BiP (BD Biosciences, 610979), α-PDIA1 (Proteintech, 11245–1-AP), α-PDIA4 (Proteintech, 14712–1-AP) α-PDIA6 (Proteintech, 18233–1-AP), α-proinsulin (HyTest Ltd., 2PR8, CCI-17). Guinea pig polyclonal α-insulin antibody was produced in-house. For secondary antibodies, goat α-mouse, goat α-rabbit, and donkey α-guinea pig antibodies were used in 1:5000 (Li-Cor, IRDye−800CW or IRDye−680RD).

## Pancreas tissue Transmission Electron Microscopy

Samples were prepared according to the UCSD Cellular and Molecular Medicine Electron Microscopy Facility protocols. Mouse pancreata were perfused in modified Karnovsky's fixative (2.5% glutaraldehyde and 2% paraformaldehyde (PFA) in 0.15M sodium cacodylate buffer, pH 7.4) and fixed for at least 4 hr, post-fixed in 1% osmium tetroxide in 0.15M cacodylate buffer for 1 hr and stained *en bloc* in 2% uranyl acetate for 1 hr. Samples were dehydrated in ethanol, embedded in Durcupan epoxy resin (Sigma-Aldrich, Inc St. Louis), sectioned at 50 to 60 nm on a Leica UCT ultramicrotome, and delivered to Formvar and carbon-coated copper grids. Sections were stained with 2% uranyl acetate for 5 min and Sato's lead stain (*Sato, 1968*) for 1 min. Images were obtained using a Tecnai G2 Spirit BioTWIN transmission electron microscope equipped with an Eagle 4 k HS digital camera (FEI, Hilsboro, OR) with indicated magnifications. Insulin granule numbers were counted manually on images taken at 4800X magnification and divided by islet area measured by image J software. Insulin mature granule vesicles and dense core sizes were also measured by image J software.

## Pancreas immunohistochemistry

Pancreata were harvested and fixed in 4% PFA. Paraffin embedding, sectioning, and slide preparations were done in the SBP Histopathology Core Facility. Sections were stained with the following antibodies; α-glucagon (Abcam, K79bB10), α-PDIA1 (Proteintech, 11245–1-AP), α-proinsulin (HyTest Ltd., 2PR8, CCI-17), and DAPI (Fisher Scientific). Guinea pig polyclonal α-insulin antibody was produced in-house. For secondary antibodies, Alexa Fluor 488 goat α-rabbit IgG, Alexa Fluor 488 goat α-mouse IgG, Alexa Fluor 594 goat α-mouse IgG, and Alexa Fluor 594 goat α-guinea pig IgG antibodies were used (Invitrogen). Images were taken by Zeiss LSM 710 confocal microscope with a 40X objective lens. Scale bar, 20 µm.

For ß cell area measurement, pancreata were harvested, fixed in 4% PFA and embedded in paraffin. Three sections were prepared at 200 µm intervals for each pancreas and stained with guinea pig polyclonal insulin antibody and DAPI. Alexa Fluor 594 goat α-guinea pig IgG was used as a secondary antibody. Images were taken by an Aperio FL Scanner (Leica). Insulin stained ß cell area, islet area, and pancreas area were measured by Aperio Imagescope software.

## Analysis of oxidative stress

Islets were plated onto CellCarrier-96 ultra microplates (Perkin Elmer) in phenol red-free RPMI 1640 medium supplemented with 10% FBS, 1% penicillin/streptomycin, 100 µg/ml primocin, 10 mM Hepes, and 1 mM sodium pyruvate one day prior to staining. Islets were treated with menadione (AdipoGen Life Science, 10 µM) for 3 hr at 37°C. Islets were stained with CellROX Deep Red reagent (Molecular Probes, C10422, 100 µM) and Hoechst 33342 (Invitrogen, 10 µg/ml) for 1 hr at the same time. Islets were washed three times with HBSS and incubated in HBSS for imaging while temperature and $CO_2$ were controlled. Images were obtained by an Opera Phenix high content screening system (63X objective lens) in the SBP High Content Screening (HCS) Facility and seven z-stack images (1 µm interval) were combined.

## Statistical analysis

Data are indicated as Mean ± SEM. Statistical significance was evaluated by unpaired two-tailed Student's t test. P-values are presented as *$p < 0.05$, **$p < 0.01$, ***$p < 0.001$.

## Acknowledgements

We thank the UCSD EM and microscopy cores and the SBP Histopathology Core. Dr. J Cho graciously provided the floxed *Pdia1* mice (U Illinois, Chicago, IL). Dr. RL Wiseman (The Scripps Research Institute, La Jolla, CA) kindly provided the PDIA1 inhibitor KSC-34. We thank Kimberly

Nagle for assistance with preparation of the manuscript. Portions of this work were supported by NIH/NCI Grants R01DK113171 (RJK), R01DK103185 (RJK), R24DK110973 (RJK, PA, PIA) and the SBP NCI Cancer Center Grant P30 CA030199. RJK is a member of the UCSD DRC (P30 DK063491) and Adjunct Professor in the Department of Pharmacology, UCSD.

## Additional information

### Funding

| Funder | Grant reference number | Author |
|---|---|---|
| National Institutes of Health | R01DK113171 | Randal J Kaufman |
| National Institutes of Health | R01DK103185 | Randal J Kaufman |
| National Institutes of Health | R24DK110973 | Pamela Itkin-Ansari<br>Peter Arvan<br>Randal J Kaufman |
| National Institutes of Health | P30 CA030199 | Randal J Kaufman |
| National Institutes of Health | P30 DK063491 | Randal J Kaufman |

The funders had no role in study design, data collection and interpretation, or the decision to submit the work for publication.

### Author contributions

Insook Jang, Conceptualization, Formal analysis, Investigation, Writing—original draft, Writing—review and editing; Anita Pottekat, Investigation, Writing—review and editing; Juthakorn Poothong, Jacqueline Lagunas-Acosta, Adriana Charbono, Investigation; Jing Yong, Formal analysis, Writing—review and editing; Zhouji Chen, Ming Liu, Conceptualization, Writing—review and editing; Donalyn L Scheuner, Conceptualization, Supervision, Writing—review and editing; Pamela Itkin-Ansari, Writing—review and editing; Peter Arvan, Conceptualization, Formal analysis, Writing—review and editing; Randal J Kaufman, Conceptualization, Supervision, Funding acquisition, Project administration, Writing—review and editing

### Author ORCIDs

Insook Jang (iD) https://orcid.org/0000-0001-7567-3849
Jing Yong (iD) https://orcid.org/0000-0002-4970-408X
Peter Arvan (iD) https://orcid.org/0000-0002-4007-8799
Randal J Kaufman (iD) https://orcid.org/0000-0003-4277-316X

### Ethics

Animal experimentation: All procedures were performed by protocols and guidelines reviewed and approved by the Institutional Animal Care and Use Committee (IACUC) at the SBP Medical Discovery Institute (AUF#. 17-066).

### Decision letter and Author response

Decision letter https://doi.org/10.7554/eLife.44528.018
Author response https://doi.org/10.7554/eLife.44528.019

## Additional files

### Supplementary files

• Transparent reporting form
DOI: https://doi.org/10.7554/eLife.44528.016

### Data availability

All data generated or analyzed during this study are included in the manuscript and supporting files.

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
