## [Decision Letter]

Thank you for submitting your work entitled "PDIA1/P4HB is required for efficient proinsulin maturation and ß cell health in response to diet induced obesity" for consideration by *eLife*. Your article has been reviewed by a Senior Editor (Vivek Malhotra), a Reviewing Editor, and three reviewers.

Our decision has been reached after consultation between the reviewers. Based on these discussions and the individual reviews below, we regret to inform you that your work will not be considered further for publication in *eLife*.

The role of protein disulfide isomerases in the folding of insulin is of great interest. Jang et al., generated mice that have a β cell-specific knockout of one of several PDIs that are expressed in islets. The impact of the knockout on blood glucose levels was relatively subtle in the absence of a high fat diet. The reviewers raised major concerns about the manuscript that are unlikely to be addressed in the time period allowed for revisions. The three reviews are included below.

Reviewer #1:

The manuscript from Jang and colleagues investigates the impact of an islet-specific knockout of the PDIA1 gene on the biosynthesis of insulin in mice. PDIA1 is the most abundant PDI family member expressed in the β cells of islets. The central finding of the manuscript is that a deficiency in PDIA1 results in a reduction in insulin secretion and an increase in the proinsulin/insulin ratio that is manifested in mice that are fed a high fat diet. The reduction in insulin secretion in the knockout mice causes an elevation of blood glucose and glucose intolerance. β cells show abnormal morphology including nuclear condensation, ER engorgement, mitochondrial swelling and decreased granule formation. A more severe impact of Pdia1 knockout is probably not observed due to the expression of other PDI family members in the knockout cells. While this manuscript is of potential interest to the readership of *eLife* there are several major weaknesses.

Essential revisions:

1) The major weakness of this manuscript is the gel analysis of the disulfide linked proinsulin complexes. The gel patterns shown in Figure 5C, Figure 5D, Figure S2A and Figure 6E show highly variable extents of disulfide-linked complexes, particularly in wild type islets. It seems possible that disulfide scrambling has occurred during sample preparation. A standard method to prevent disulfide scrambling is to treat cells with N-ethyl maleimide prior to cell lysis, to block free thiols. In Fig, 5C monomeric proinsulin can't account for more than a few% of total proinsulin in either wild type or KO cells. The proinsulin bands migrating between 14-49 kDa (dimers to pentamers?) appear to be similar in abundance to larger aggregates (49-198 kDa). In Figure 6E, monomeric insulin appears to be a higher% of total in both wild type and KO cells in the absence of Menadione. The dimer to pentamer category is minor in Figure 6E relative to the large aggregates. Quantification of relative protein levels (graphs in Figure 5C, etc.) may be useful for WT and KO comparisons, but it obscures the more serious problem that disulfide linked complexes are predominant in wild type islets. Disulfide-linked proinsulin aggregates are predominant even when wild type mice are fed a standard diet (Figure 5D). These results are inconsistent with the results/conclusions presented in the accompanying manuscript.

2) The X-axis in Figure 6D is confusing since the labels do not point at a tick mark. We realize that nuclei sizes have been binned into 10-integer ranges, but the offset between the labels and tick marks makes it unclear whether the bin with the highest frequency in the upper graph covers 35 to 45, or 40-50. It is also unclear how the mean value for nuclear size can be so severely left-shifted relative to the frequency distribution. For example, the mean value for vehicle treated cells is shown to fall in the bin labeled 30, but 90% of nuclei fall in bins with higher nuclear sizes. How can the mean of a sample not have a more obvious relationship to the frequency distribution? Is this due to an error in calculation of the mean? Perhaps this graph would be improved by using a smaller bin size.

3) The legend to Figure 7A contains excess information. This figure legend reads as if it was copied from another document that the authors have written (early draft of the introduction?), as it contains two citations and is overburdened with information that does not correlate with the figure. For example, why does the legend indicate that β cells express two Ero1 proteins, when a single ERO1 is shown in panel A? Why is there text about *Akita* diabetic mice in the legend to Figure 7A? The diagram for the disulfide-linked PI complexes is vague. Are the blue circles supposed to be other proinsulin molecules, or are these supposed to be other ER proteins with a free thiol? The impression that Figure 7 was tacked onto the manuscript as an afterthought is also apparent from the in text citations in the Discussion section. Figure 7 is mentioned twice, but the text does not indicate which panel. The citations to Figure 7 that are in the discussion could just as well cite the experimental figures as the diagram in Figure 7B isn't that informative.

Reviewer #2:

Jang et al., studied the role of protein disulfide isomerase A1 (PDIA1) in the regulation of β cell function. They generated β cell-specific Pdia1 knockout mice and found that feeding a high-fat diet (HFD) induced glucose intolerance along with hypoinsulinemia and increased proinsulin/insulin ratio compared to similarly treated wildtype mice. PDIA1 deficiency in β cells induced morphological alterations in the ER and mitochondria and impaired the maturation of secretory granules; this was associated with accumulation of misfolded proinsulin aggregates and islet vulnerability to oxidative stress. Collectively, these findings suggest that PDIA1, which is required for proinsulin maturation and insulin production, is involved in the regulation of ER redox state. This manuscript together with the accompanying paper by the Arvan group support the hypothesis that ER dysfunction with subsequent generation of misfolded proinsulin aggregates plays an important role in the pathophysiology of diabetes. However, there are several major issues that should be addressed.

Specific comments:

1) Oxidoreductases are important for disulfide bonds formation; however, the impact of PD1A1 knockout is relatively small. Thus, glucose homeostasis in young animals was normal; mice developed β cell dysfunction only after prolonged metabolic challenge and probably aging. KO mice developed modest fasting hyperglycemia even on a HFD for 5 months. In the re-feeding experiment (Figure 3D), while proinsulin/insulin ratio was increased, blood glucose was still not increased. In addition, there was no effect on the expression of proinsulin, β cell transcription factors, pancreatic convertases, oxidative stress and UPR genes (except for GRP94). What is the relative importance of PDIA1 vs all other disulphide isomerases and redox enzymes? This question should be addressed.

2) The conclusion that KO mice develop hyperglycemia during aging is based on two KO mice (Figure 2D and E). It is not possible to perform statistical comparison with such a small number of animals. The number of animals in each group should be increased. What were blood glucose levels in aged male Pdia1 KO mice compared to controls? This part of the figure should be either extended or deleted.

3) The quantifications in Figure 5B and 5C were performed on 2 WT and 3 KO mice only. Statistical comparison with such a small number of animals is not possible. Note that in Figure 5C (lane #1, WT) the amount of proinsulin complexes is much higher than the monomeric form of proinsulin and is even greater than that in KO mice. In Figure 5C, the total proinsulin level in KO islets was higher than in WT controls (reducing gel). Is there greater affinity of the proinsulin antibody to misfolded aggregates compared to native proinsulin? These findings should be explained.

4) The pulse-chase experiments showed that PDIA1 KO did not affect proinsulin conversion to insulin (Supplementary Figure 3); this is inconsistent with the authors' suggestion that PDIA1 deficiency impairs proinsulin folding and maturation.

5) It would be expected that impairment of ER redox state by deleting PDIA1 generates oxidative stress in response to metabolic challenge; however, ROS production was similar in wildtype and KO mice on HFD (Figure 7A). ROS production along with nuclear condensation was increased in KO mice only after treatment with a potent pro-oxidant. This raises concerns regarding the importance of PDIA1 in the regulation of ER homeostasis in β cells.

Reviewer #3:

The manuscript by Jang et al., describes studies on the β cell-specific knockout of Pdia1 gene and its effects on metabolic homeostasis and proinsulin processing. The authors used an Ins-CreER transgene to inducibly delete Pdia1 in β cells, then fed mice and their littermate controls a 45% HFD to increase cellular demand on insulin production. They observed glucose intolerance/diabetes, elevated proinsulin/insulin ratios, and evidence of dysmorphic insulin granules in KO mice. The authors conclude that the absence of a critical oxidoreductase to properly guide disulfide bond formation results in an ER stress-like state that leads to β cell dysfunction. The study is of clear importance to the β cell field and emphasizes the critical role of the ER and proinsulin folding in β cell function and glucose homeostasis; it furthers the narrative advanced by the seminal work of this group over the years. However, there are two major issues, and two moderate ones that require further experimentation and consideration:

Major comments:

1) Perhaps the biggest concern with the study is the controls against which the KOs are compared. The legend to Figure 1 states that the "WT" mice are control littermates with one or two floxed Pdia1 alleles. Technically, these are not WT mice, but that is a minor concern that can be easily corrected. The more significant concern is that no Cre+ controls were used. As the authors are aware, particularly with the RIP transgene, Cre+ mice can exhibit glucose intolerance due to β cell stress induced by the transgene. In the absence of this control, the overall conclusions of the study remain uninterpretable, since only the KO mice bear this transgene. I am not arguing that the results of this study are invalid in the absence of the controls, but the magnitude of the effects might not be as great as the authors purport.

2) The study is that it takes a somewhat biased perspective from start to finish. The authors chose this gene for study based on their understanding of its role in ER protein folding. As such, many of the studies presented in the manuscript are leveraged to support this bias. It would have been more convincing if the authors had performed unbiased RNA-Seq studies from the islets, rather than targeted gene RT-PCR. What might be happening to β cell differentiation genes? Or to a more comprehensive list of other oxidoreductases? Or to genes involved in Ca regulation (e.g. SERCA, Ryanodyne R, etc.)? Oxidative stress pathways? A more comprehensive bioinformatics pathway analysis (e.g. Gene Ontology) could address these questions and provide more unbiased evidence to support the authors' hypothesis. Of course, the comparator would have to be done against Cre+ controls, as noted above.

3) The RIP-Cre transgene is known to be expressed in the brain (hypothalamus), and this could affect glucose homeostasis, as has been noted previously. The authors should probably test the mRNA levels of Pdia1 in hypothalamus to confirm or refute this contention. Therefore, the use of Cre+ controls as noted above becomes especially more important.

4) The authors state that UPR genes are not significantly altered, but that there is ER stress occurring in the β cells (based both on the increase in BiP and the morphology in the EM images). Granted that the UPR is a response to ER stress, but does this disconnect refer to a defective UPR or simply that the UPR has failed at this stage of analysis and ER stress predominates?

---

## [Author Response]

Reviewer #1:The manuscript from Jang and colleagues investigates the impact of an islet-specific knockout of the PDIA1 gene on the biosynthesis of insulin in mice. PDIA1 is the most abundant PDI family member expressed in the β cells of islets. The central finding of the manuscript is that a deficiency in PDIA1 results in a reduction in insulin secretion and an increase in the proinsulin/insulin ratio that is manifested in mice that are fed a high fat diet. The reduction in insulin secretion in the knockout mice causes an elevation of blood glucose and glucose intolerance. β cells show abnormal morphology including nuclear condensation, ER engorgement, mitochondrial swelling and decreased granule formation. A more severe impact of Pdia1 knockout is probably not observed due to the expression of other PDI family members in the knockout cells. While this manuscript is of potential interest to the readership of eLife there are several major weaknesses.Essential revisions:1) The major weakness of this manuscript is the gel analysis of the disulfide linked proinsulin complexes. The gel patterns shown in Figure 5C, Figure 5D, Figure S2A and Figure 6E show highly variable extents of disulfide-linked complexes, particularly in wild type islets. It seems possible that disulfide scrambling has occurred during sample preparation. A standard method to prevent disulfide scrambling is to treat cells with N-ethyl maleimide prior to cell lysis, to block free thiols. In Fig, 5C monomeric proinsulin can't account for more than a few% of total proinsulin in either wild type or KO cells. The proinsulin bands migrating between 14-49 kDa (dimers to pentamers?) appear to be similar in abundance to larger aggregates (49-198 kDa). In Figure 6E, monomeric insulin appears to be a higher% of total in both wild type and KO cells in the absence of Menadione. The dimer to pentamer category is minor in Figure 6E relative to the large aggregates. Quantification of relative protein levels (graphs in 5C, etc.) may be useful for WT and KO comparisons, but it obscures the more serious problem that disulfide linked complexes are predominant in wild type islets. Disulfide-linked proinsulin aggregates are predominant even when wild type mice are fed a standard diet (Figure 5D). These results are inconsistent with the results/conclusions presented in the accompanying manuscript.

First, we were aware of this variability and devoted considerable effort to optimize the analysis to obtain more reproducible non-reducing gels. As the reviewer suggested we have treated islets with NEM prior to lysis and demonstrated that alkylation did not significantly alter the pattern of oligomers and HMW complexes, although NEM treatment did result in a relative decrease in homo-oligomers of proinsulin (mw 14-49 kDa) with a corresponding increase in HMW complexes (shown in New Figure 5E). One possible interpretation of the findings is that PDIA1 is required to reduce HMW proinsulin complexes and NEM treatment inactivates PDIA1, thereby stabilizing the HMW complexes. We also provide independent evidence in support of this conclusion, using a PDI-specific inhibitor (see below).

The reviewer was also concerned that we detected HMW complexes in wildtype islets. Indeed, we do detect disulfide-linked oligomers in wildtype islets. The major conclusion from the accompanying paper by Arungagiri et al., is that oligomers are detected in healthy human islets (Figure 3A,B), in wildtype murine islets and increase with increased proinsulin synthesis (Figure 3C) and are detected in *db/db* islets prior to onset of hyperglycemia (Figure 5C). The ability to detect these disulfide-linked complexes in wildtype and healthy islets is due to epitope exposure that is not detected with our polyclonal anti-insulin antibodies. As we previously demonstrated, addition of DTT to islets in vitro collapsed the multimers down to monomeric proinsulin and an apparently resistant dimer migrating at 16 kDa (New Figure 5F). Importantly, treating the non-reducing gel with DTT prior to transfer greatly increases detection of the monomeric proinsulin indicating that epitope exposure to this antibody increases upon proinsulin reduction (New Figure 5D,E). The increase in HMW complexes is also evident in PDIA1-deficient β cells and even more so when the islets are exposed to the oxidant menadione (New Figure 6F). Analysis of these results also demonstrates significant reproducibility of the data from islet preparations from 6 individual mice (Figure 5C,D).

The corresponding description is in subsection “KO islets have an increased intracellular proinsulin/insulin ratio with accumulation of high molecular weight (HMW) proinsulin complexes” and subsection “Oxidant treatment of Pdia1 KO islets increases accumulation of HMW proinsulin complexes”.

2) The X-axis in Figure 6D is confusing since the labels do not point at a tick mark. We realize that nuclei sizes have been binned into 10-integer ranges, but the offset between the labels and tick marks makes it unclear whether the bin with the highest frequency in the upper graph covers 35 to 45, or 40-50. It is also unclear how the mean value for nuclear size can be so severely left-shifted relative to the frequency distribution. For example, the mean value for vehicle treated cells is shown to fall in the bin labeled 30, but 90% of nuclei fall in bins with higher nuclear sizes. How can the mean of a sample not have a more obvious relationship to the frequency distribution? Is this due to an error in calculation of the mean? Perhaps this graph would be improved by using a smaller bin size.

We thank the reviewer for this constructive suggestion. We have redrawn the Figure and placed the ‘tik’ marks to more accurately reflect the different bins. We agree that inclusion of the mean nuclear areas from Figure 6C into the graphs in Figure 6D generated confusion and they have now been removed.

3) The legend to Figure 7A contains excess information. This figure legend reads as if it was copied from another document that the authors have written (early draft of the introduction?), as it contains two citations and is overburdened with information that does not correlate with the figure. For example, why does the legend indicate that β cells express two Ero1 proteins, when a single ERO1 is shown in panel A? Why is there text about Akita diabetic mice in the legend to Figure 7A? The diagram for the disulfide-linked π complexes is vague. Are the blue circles supposed to be other proinsulin molecules, or are these supposed to be other ER proteins with a free thiol? The impression that Figure 7 was tacked onto the manuscript as an afterthought is also apparent from the in text citations in the discussion. Figure 7 is mentioned twice, but the text does not indicate which panel. The citations to Figure 7 that are in the discussion could just as well cite the experimental figures as the diagram in Figure 7B isn't that informative.

Thank you for picking up on this! Indeed, the legend to Figure 7 was inadvertently included from an earlier draft of the manuscript and we have now deleted it and revised the model. Specifically, we have omitted Figure 7B and only focus on the points in Figure 7A that are most relevant to our findings in this manuscript regarding the pathway of proinsulin oxidation, disulfide-linked oligomer and HMW complex formation. Our results provide new information suggesting a role of PDIA1 in the inter-conversion of oligomeric, HMW proinsulin complexes and native folded proinsulin.

Reviewer #2:Jang et al., studied the role of protein disulfide isomerase A1 (PDIA1) in the regulation of β cell function. They generated β cell-specific Pdia1 knockout mice and found that feeding a high-fat diet (HFD) induced glucose intolerance along with hypoinsulinemia and increased proinsulin/insulin ratio compared to similarly treated wildtype mice. PDIA1 deficiency in β cells induced morphological alterations in the ER and mitochondria and impaired the maturation of secretory granules; this was associated with accumulation of misfolded proinsulin aggregates and islet vulnerability to oxidative stress. Collectively, these findings suggest that PDIA1, which is required for proinsulin maturation and insulin production, is involved in the regulation of ER redox state. This manuscript together with the accompanying paper by the Arvan group support the hypothesis that ER dysfunction with subsequent generation of misfolded proinsulin aggregates plays an important role in the pathophysiology of diabetes. However, there are several major issues that should be addressed.Specific comments:1) Oxidoreductases are important for disulfide bonds formation; however, the impact of PD1A1 knockout is relatively small. Thus, glucose homeostasis in young animals was normal; mice developed β cell dysfunction only after prolonged metabolic challenge and probably aging. KO mice developed modest fasting hyperglycemia even on a HFD for 5 months. In the re-feeding experiment (Figure 3D), while proinsulin/insulin ratio was increased, blood glucose was still not increased. In addition, there was no effect on the expression of proinsulin, β cell transcription factors, pancreatic convertases, oxidative stress and UPR genes (except for GRP94). What is the relative importance of PDIA1 vs all other disulphide isomerases and redox enzymes? This question should be addressed.

This reviewer brings up an important point as to what is the role of other oxidoreductases. We are performing such experiments as part of a future study, but to bolster our conclusions in the current study of PDIA1, we have added new data with KSC-34, an inhibitor that selectively interacts with the A catalytic site of PDIA1 (Cole et al., 2018. We show that KSC-34 increases disulfide-linked HMW proinsulin complexes in islets and also renders the islets susceptible to the oxidant Menadione, with a marked further increase in HMW disulfide-linked proinsulin-containing complexes (New Figure 6—figure supplement 1). These behaviors phenocopy the effect of Pdia1 deletion. These data provide both pharmacological and genetic support for our findings which indicate that PDIA1 facilitates proinsulin maturation by minimizing the accumulation of HMW disulfide linked proinsulin-containing complexes. This is especially important upon metabolic challenge, such as high-fat diet. The corresponding description is in subsection “Pharmacological inhibition of PDIA1 recapitulates effects of Pdia1 gene deletion”.

Regarding the fast-refed blood glucose, it is not surprising there is no change in refed glucose levels. These young mice are likely more insulin sensitive and there are many additional factors that maintain blood glucose within a narrow range. This was pointed out by Jim Johnson in studies of Proinsulin gene deletion that show after 12 days of proinsulin gene deletion there was no change in blood glucose (Szabat et al., 2016). To eliminate confusion, we have deleted these data because the findings are not relevant to our conclusions.

2) The conclusion that KO mice develop hyperglycemia during aging is based on two KO mice (Figure 2D and E). It is not possible to perform statistical comparison with such a small number of animals. The number of animals in each group should be increased. What were blood glucose levels in aged male Pdia1 KO mice compared to controls? This part of the figure should be either extended or deleted.

We agree with this reviewer that an n=2 (~350 day old females) cannot be used for statistics. Therefore, we removed Figure 2D,E. We have now analyzed an additional n=12 ~250 day old male mice fed normal chow and observed a statistically significant worsening of glucose tolerance in the old *Pdia1*-null mice supporting our original observation (New Figure 2D,E). At this point we cannot comment on whether the phenotype is associated with aging or due to metabolic consequences of age, which is consistent with the HFD fed mice. The corresponding description is in subsection “Generation of ß cell-specific *Pdia1* deleted mice.”

3) The quantifications in Figure 5B and 5C were performed on 2 WT and 3 KO mice only. Statistical comparison with such a small number of animals is not possible. Note that in Figure 5C (lane #1, WT) the amount of proinsulin complexes is much higher than the monomeric form of proinsulin and is even greater than that in KO mice. In Figure 5C, the total proinsulin level in KO islets was higher than in WT controls (reducing gel). Is there greater affinity of the proinsulin antibody to misfolded aggregates compared to native proinsulin? These findings should be explained.

The original data in Figure 5A and C from n=2 and 3 mice has now been increased to 6 mice (3 WT and 3 KO) as biological replicates in Figures 5A (reduced), 5C (no DDT added to gel) and 5D (DTT added to gel). The results provide for significant statistical analysis (Figure 5B and 5D) and demonstrate reproducible trends between murine genotypes. The corresponding description is in subsection “KO islets have an increased intracellular proinsulin/insulin ratio with accumulation of high molecular weight (HMW) proinsulin complexes”.

Indeed, as this insightful reviewer suggested, the source for variation between monomeric and multimeric proinsulin, is that reduction of the sample greatly increases epitope exposure of monomeric proinsulin to the antibody. We didn't emphasize this issue in our study because the 2D gel analysis in the companion Arunagiri et al., paper shows that all bands in these complexes contain proinsulin. This key point is most evident by an experiment where we incubated the gel in 25mM DTT before transfer to nitrocellulose. (In addition, we now run 12% polyacrylamide gels as opposed to gradient gels to as much as possible equalize transfer conditions throughout the gel in order to obtain Westerns with greatly increased the signal for the proinsulin monomer relative to the multimers, in a more reproducible manner, as is evident in Figure 1C and Figures 5A,C,D).

Second, this reviewer was concerned about the detection of proinsulin complexes in WT islets. The major conclusion from the companion paper by Arunagiri et al., is that oligomers are detected in healthy human islets (Figure 3A,B), in wildtype murine islets, and further increase with increased proinsulin synthesis (Figure 3C) and are detected in *db/db* islets prior to onset of hyperglycemia (Figure 5C).

4) The pulse-chase experiments showed that PDIA1 KO did not affect proinsulin conversion to insulin (Supplementary Figure 3); this is inconsistent with the authors' suggestion that PDIA1 deficiency impairs proinsulin folding and maturation.

We agree with this reviewer that the pulse-chase studies did not provide significant insight into the defect in the *Pdia*1-null islets; thus, we have omitted this figure. Instead, we now include analysis of brefeldin A treatment which increases retrograde trafficking of proinsulin from the cis-Golgi back to the ER. BFA-treated WT islets significantly increased the multimeric forms of proinsulin (New Figure 6—figure supplement 1), suggesting proinsulin accumulation in the ER increases HMW complex formation. The corresponding description is on page 9 under “Proinsulin accumulation in the ER increases oligomeric and HMW complexes”.

5) It would be expected that impairment of ER redox state by deleting PDIA1 generates oxidative stress in response to metabolic challenge; however, ROS production was similar in wildtype and KO mice on HFD (Figure 7A). ROS production along with nuclear condensation was increased in KO mice only after treatment with a potent pro-oxidant. This raises concerns regarding the importance of PDIA1 in the regulation of ER homeostasis in β cells.

The reviewer is correct that we did not see increased ROS production, and thus the reviewer is also correct that our data indicate that the role of PDIA1 in the ER homeostasis of β cells does not appear to be primarily through the consumption of ROS. In our view, the central finding of our paper is that *Pdia1* deletion renders β cells less resilient to metabolic challenge because this perturbation of ER homeostasis renders proinsulin more susceptible to formation/accumulation of high molecular weight disulfide-linked complexes, resulting in downstream consequences that include diminished islet insulin. This more aptly describes the importance of PDIA1 in the regulation of ER homeostasis in β cells.

Reviewer #3:The manuscript by Jang et al., describes studies on the β cell-specific knockout of Pdia1 gene and its effects on metabolic homeostasis and proinsulin processing. The authors used an Ins-CreER transgene to inducibly delete Pdia1 in β cells, then fed mice and their littermate controls a 45% HFD to increase cellular demand on insulin production. They observed glucose intolerance/diabetes, elevated proinsulin/insulin ratios, and evidence of dysmorphic insulin granules in KO mice. The authors conclude that the absence of a critical oxidoreductase to properly guide disulfide bond formation results in an ER stress-like state that leads to β cell dysfunction. The study is of clear importance to the β cell field and emphasizes the critical role of the ER and proinsulin folding in β cell function and glucose homeostasis; it furthers the narrative advanced by the seminal work of this group over the years. However, there are two major issues, and two moderate ones that require further experimentation and consideration:Major comments:1) Perhaps the biggest concern with the study is the controls against which the KOs are compared. The legend to Figure 1 states that the "WT" mice are control littermates with one or two floxed Pdia1 alleles. Technically, these are not WT mice, but that is a minor concern that can be easily corrected. The more significant concern is that no Cre+ controls were used. As the authors are aware, particularly with the RIP transgene, Cre+ mice can exhibit glucose intolerance due to β cell stress induced by the transgene. In the absence of this control, the overall conclusions of the study remain uninterpretable, since only the KO mice bear this transgene. I am not arguing that the results of this study are invalid in the absence of the controls, but the magnitude of the effects might not be as great as the authors purport.

This reviewer makes the important point that we have not studied the impact of the *RIP-Cre^ERT^* allele in β cells. We actually did perform these studies which demonstrate the *RIP-Cre^ERT^* allele does not impact glucose homeostasis in mice with one floxed and one wt *Pdia1* allele, but neglected to include this data in the original manuscript. We now include the control without *RIP-Cre^ERT^*which is similar to *RIP-Cre^ERT^*-containing mice (New Figure 1—figure supplement 1). The corresponding description is insubsection “*Generation of ß cell-specific Pdia1 deleted mice”*. Therefore, although the *RIP-Cre^ERT^* allele may impact β cell function, it does not significantly contribute to the phenotype we have characterized.

Finally, we respect the reviewer’s concern that characterizing the genotypes simply as WT and KO did not provide sufficient information regarding specifics of the genotype. Therefore, we have changed all WT and KO labelings to reflect genotypes in the figures.

We now label all mice with respect to the genotype of WT *Pdia1*, floxed *Pdia1* and *Cre^ERT^*status as follows:

*fl/fl:Cre^ERT^* with Tam, equivalent to KO in the previous draft.

*fl/+* and *fe/fe* without *Cre^ERT^*, equivalent to WT.

*fl/+* and *fl/-* with *Cre^ERT^* and Tam, equivalent to het or WT.

2) The study is that it takes a somewhat biased perspective from start to finish. The authors chose this gene for study based on their understanding of its role in ER protein folding. As such, many of the studies presented in the manuscript are leveraged to support this bias. It would have been more convincing if the authors had performed unbiased RNA-Seq studies from the islets, rather than targeted gene RT-PCR. What might be happening to β cell differentiation genes? Or to a more comprehensive list of other oxidoreductases? Or to genes involved in Ca regulation (e.g. SERCA, Ryanodyne R, etc.)? Oxidative stress pathways? A more comprehensive bioinformatics pathway analysis (e.g. Gene Ontology) could address these questions and provide more unbiased evidence to support the authors' hypothesis. Of course, the comparator would have to be done against Cre+ controls, as noted above.

Thank you for the suggestion. As the reviewer may have surmised, we are currently working on an unbiased proteomics characteriziation of both ER oxidoreductases and proinsulin interactor proteins in human islets, so we very much intend to follow up on the reviewer’s point involving omics-based screening. However, that is quite a major undertaking and an entire project (and manuscript) unto itself. In the current manuscript, we think it is unlikely that mRNA-Seq would change our primary conclusion that *Pdia1* deletion renders β cells less resilient to metabolic challenge because this perturbation renders proinsulin more susceptible to formation/accumulation of high molecular weight disulfide-linked complexes, resulting in downstream consequences that include diminished islet insulin.

However, in deference to the reviewer’s concern, we have now included additional qRT-PCR results for PDIA1, PDIA3, PDIA4, PDIA6 and Serca2B in the New Figure 3G. The corresponding description is on page 6 under “Pdia1 is not required for expression of ß cell-specific genes, antioxidant response genes or cell death genes”.

3) The RIP-Cre transgene is known to be expressed in the brain (hypothalamus), and this could affect glucose homeostasis, as has been noted previously. The authors should probably test the mRNA levels of Pdia1 in hypothalamus to confirm or refute this contention. Therefore, the use of Cre+ controls as noted abovebecomes especially more important.

Indeed, another issue regarding RIP-Cre^ERT^ is potential hypothalamic expression. We now show that WT and Pdia1 KO mice (n=6/genotype) express similar levels of PDIA1 by Western blotting of hypothalamic tissue (New Figure 1—figure supplement 1B). The animals also show similar body weights (Figure 1A). Therefore, although the RIP-Cre^ERT^ allele may impact β cell function, it does not significantly contribute to the phenotype we have characterized. The corresponding description is in subsection “Generation of ß cell-specific *Pdia1* deleted mice.”

In addition, we previously reported (Hassler et al., 2015) there was no difference between IRE1α WT and KO mice in serum dopamine, which is synthesized in the arcuate nucleus of the hypothalamus. Although we did detect Cre positive staining in brain sections in the Hassler study, there was little difference in growth hormone-releasing hormone (GHRH) expression between the KO mice and the controls.

4) The authors state that UPR genes are not significantly altered, but that there is ER stress occurring in the β cells (based both on the increase in BiP and the morphology in the EM images). Granted that the UPR is a response to ER stress, but does this disconnect refer to a defective UPR or simply that the UPR has failed at this stage of analysis and ER stress predominates?

We thank the reviewer for raising this point. Our new analysis of a greater number of islet preps from 3 KO and 3 WT mice on HFD now demonstrate significant increases in several UPR-induced proteins; BiP, PDIA4 and PDIA6, which is more consistent with the ER morphology. We have now stated this more clearly in the manuscript – thanks! The corresponding description is in subsection “KO islets have an increased intracellular proinsulin/insulin ratio with accumulation of high molecular weight (HMW) proinsulin complexes”.